# CoEmoGen: Towards Semantically-Coherent and Scalable Emotional Image Content Generation

**Kaishen Yuan**[1*]**, Yuting Zhang**[1*]**, Shang Gao**[1]**, Yijie Zhu**[2]**, Wenshuo Chen**[1]**, Yutao Yue**[1,3†]

[1] The Hong Kong University of Science and Technology (Guangzhou)
[2] Harbin Institute of Technology (Shenzhen)
[3] Institute of Deep Perception Technology, JITRI
{kyuan156,yzhang430}@connect.hkust-gz.edu.cn
yutaoyue@hkust-gz.edu.cn

## Abstract

Emotional Image Content Generation (EICG) aims to generate semantically clear and emotionally faithful images based on given emotion categories, with broad application prospects. While recent text-to-image diffusion models excel at generating concrete concepts, they struggle with the complexity of abstract emotions. There have also emerged methods specifically designed for EICG, but they excessively rely on word-level attribute labels for guidance, which suffer from semantic incoherence, ambiguity, and limited scalability. To address these challenges, we propose CoEmoGen, a novel pipeline notable for its semantic coherence and high scalability. Specifically, leveraging multimodal large language models (MLLMs), we construct high-quality captions focused on emotion-triggering content for context-rich semantic guidance. Furthermore, inspired by psychological insights, we design a Hierarchical Low-Rank Adaptation (HiLoRA) module to cohesively model both polarity-shared low-level features and emotion-specific high-level semantics. Extensive experiments demonstrate CoEmoGen's superiority in emotional faithfulness and semantic coherence from quantitative, qualitative, and user study perspectives. To intuitively showcase scalability, we curate EmoArt, a large-scale dataset of emotionally evocative artistic images, providing endless inspiration for emotion-driven artistic creation. The dataset and code are available at https://github.com/yuankaishen2001/CoEmoGen.

## 1 Introduction

> *"The artist is a receptacle for emotions that come from all over the place: from the sky, from the earth, from a scrap of paper, from a passing shape..."*   –Pablo Picasso

Emotion is an innate human instinct that shapes our perceptions and reactions to the world (Minsky, 2007), and to better enable artificial intelligence to understand and respond to human emotional needs, affective computing has rapidly advanced in recent years (Tao & Tan, 2005), with Visual Emotion Analysis (VEA) emerging as a hot research area. VEA explores the emotional information embedded in visual stimuli and analyzes human responses, offering significant potential for real-world applications such as mental health (Wieser et al., 2012; Li et al., 2024; 2025a) and more. As the field progresses, an intriguing question arises: can we reverse the VEA paradigm, shifting from emotional recognition to generating images that evoke specific emotions, thereby enabling emotionally intelligent content creation?

Owing to the remarkable advancements in diffusion models (Ho & Salimans, 2022; Dhariwal & Nichol, 2021; Song et al., 2020; Ho et al., 2020), a large number of text-to-image models have emerged, enabling users to input prompts and control conditions to generate high-quality customized images (Zhang et al., 2023; Peebles & Xie, 2023; Ruiz et al., 2023; Rombach et al., 2022; Tian et al.,

---

[*] Equal contribution
[†] Corresponding author

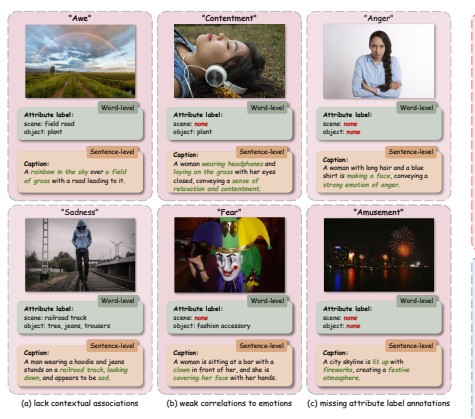

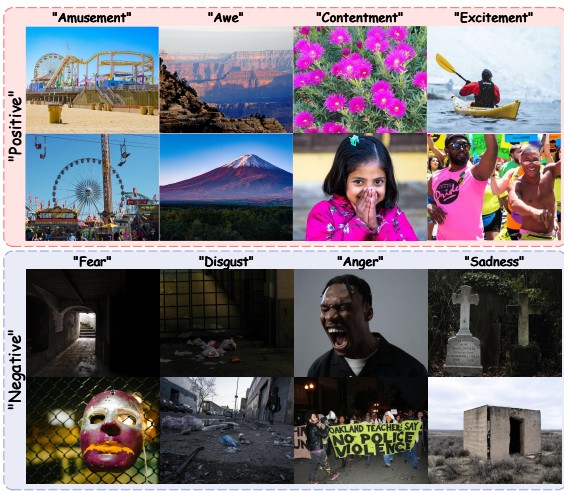

Figure 1: Comparison of sentence-level captions and word-level attribute labels as semantic guidance, where the latter suffers from (a) lack of contextual association, (b) weak correlation to emotions, and (c) missing annotations.

Figure 2: Emotions of the same polarity exhibit similarity in low-level features but differ in high-level fine-grained semantics.

2025; Gal et al., 2022). Unfortunately, although these existing models are proficient at generating *concrete* concepts (*e.g., dogs, tables, cars*), they struggle and even collapse when tasked with generating *abstract* concepts such as emotions (*e.g., contentment, awe, sadness*) (Yang et al., 2024). This limitation hampers the progress of emotional intelligence, highlighting the urgent need for the exploration of a model capable of generating images that evoke specific emotions.

Some attempts have sought to achieve target emotion conveyance by modifying the color or style of images, but their potential is constrained by fixed image content (Liu et al., 2018; Weng et al., 2023). From a psychological perspective, visual emotions are strongly correlated with specific semantics (Brosch et al., 2010), making mere adjustments to low-level features ineffective in conveying the intended emotion. To bridge the gap between abstract emotions and concrete images, EmoGen (Yang et al., 2024) pioneered the definition of Emotional Image Content Generation (EICG) as generating semantically clear and emotionally faithful visual content based on a given emotion, and explored leveraging word-level attribute labels (*i.e., objects or scenes*) from EmoSet (Yang et al., 2023) as guidance to align the constructed emotion space with the semantically rich Contrastive Language-Image Pre-training (CLIP) (Radford et al., 2021) space, making a promising step toward effective emotional image generation. Nevertheless, overly relying on attribute labels poses the following limitations: (1) Restricted to word-level and used in isolation, attribute labels lack contextual connections, which prevents them from conveying comprehensive semantics (Figure 1 (a)). (2) Some annotated attribute labels may weakly correlate with emotions, while the truly emotion-triggering elements are missing, leading to ambiguity (Figure 1 (b)). (3) Attribute labels in EmoSet are not always annotated, which significantly limits the diversity and scalability of training corpus (Figure 1 (c)). Therefore, exploring ways to overcome these limitations and achieve greater flexibility in EICG is a worthwhile direction for research.

Based on the above observations, we propose CoEmoGen, a novel pipeline towards semantically coherent and highly scalable EICG. Specifically, we focus on sentence-level captions rather than word-level attribute labels for semantically-coherent guidance, integrating visual-emotional logic through rich context and achieving alignment closer to human cognition. Leveraging the powerful capabilities of multimodal large language models (MLLMs) (Li et al., 2025c), we organize effective captions that are concise yet focused on emotion-related content for the images in EmoSet, and use CLIP space for filtering to mitigate the noise inevitably introduced by MLLM hallucinations (Bai et al., 2024). Figure 1 shows the superiority of sentence-level semantically-coherent guidance over word-level guidance, while the aforementioned standardized construction paradigm also lays the foundation for high scalability. Furthermore, inspired by the psychological observation that emotions of the same polarity (*positive or negative*) share similarities in low-level visual attributes (*e.g., brightness, color*) but differ in high-level semantic features (as illustrated in Figure 2) (Yang et al., 2023), we propose a Hierarchical Low-Rank Adaptation (HiLoRA) module, which includes polarity-shared LoRAs to capture common base features within the same polar-

ity and emotion-specific LoRAs to learn the fine-grained exclusive elements of specific emotions. Extensive experiments, including quantitative comparisons, qualitative analyses, and a user study, demonstrate the superiority of CoEmoGen in semantic coherence and emotional fidelity. To further showcase its scalability, we collect EmoArt, a dataset of 13,633 emotionally evocative artistic images, which CoEmoGen effortlessly leverages to inspire artists in creating emotion-driven artworks. Our contributions can be summarized as follows:

- We propose CoEmoGen, a novel semantically-coherent and highly scalable EICG pipeline.

- We introduce reliable captions with emotion-related content for EmoSet, leveraging their rich coherent context for sentence-level semantic guidance.

- Inspired by psychology, we design a HiLoRA module with polarity-shared LoRAs for modeling the common base features and emotion-specific LoRAs for capturing the fine-grained exclusive elements.

- Extensive experiments, encompassing quantitative, qualitative, and a user study, showcase CoEmoGen's superiority in EICG, while EmoArt, a dataset composed of emotionally stimulating artworks, is constructed to further highlight its flexible scalability.

## 2 RELATED WORK

**Visual emotion analysis.** Visual Emotion Analysis (VEA) aims to predict people's emotional responses to visual stimuli and has been studied for over two decades (Zhao et al., 2021; Mohamed et al., 2022; Achlioptas et al., 2021). The most commonly employed Mikels model classifies emotions into eight categories: *amusement, awe, contentment, excitement, anger, disgust, fear,* and *sadness*, with the first four belonging to the *positive* polarity and the latter four to the *negative* polarity (Mikels et al., 2005). In the early stages, inspired by psychology and art theory, Machajdik & Hanbury (2010) extracted specific image features related to emotions, such as color and composition, for analysis. With the rapid development of deep learning, Rao et al. (2020) proposed MldrNet, which integrates different levels of deep representations (image semantics, aesthetics, and low-level features) to classify emotions across various types of images. Yang et al. (2021) first introduced stimulus selection, extracting unique emotional features from different stimuli. Ultimately, VEA is a classification task, while reversing this process to generate images that evoke specific emotions has broad practical applications in areas such as mental health and artistic creation, making it worth further exploration.

**Emotional image content generation.** Recently, with the remarkable progress of diffusion models (Ho & Salimans, 2022; Ho et al., 2020), a large number of text-to-image models have emerged, allowing users to generate high-quality and diverse images (Peebles & Xie, 2023; Betker et al., 2023; Rombach et al., 2022; Tian et al., 2025). Furthermore, to achieve customization, several personalized text-to-image generation methods (Zhang et al., 2023; Ruiz et al., 2023; Zhu et al., 2024) have been proposed, which typically rely on learning new embeddings or fine-tuning model parameters. While these methods have attained impressive results in generating concrete concepts, they fall short when it comes to abstract concepts such as emotions. Some attempts aim to modify low-level visual elements to alter the emotional tone of input image, thus performing image emotion transfer. Representative works include Peng *et al.*'s changing the emotions evoked by adjusting the image's color and texture features (Peng et al., 2015), and Sun *et al.*'s proposal to activate a specified emotion through style transfer (Sun et al., 2023). Due to the fixed content of the image, and the strong correlation between visual emotions and specific semantics, simply modifying these superficial features is insufficient to evoke the desired emotion. Therefore, EmoGen (Yang et al., 2024) initially defined the EICG task as creating semantically clear and emotionally faithful visual content based on a specified emotion category, and it achieved promising outcomes in generating emotionally evocative images. Nevertheless, due to the reliance on word-level attribute labels, which lack semantic context, are weakly related to emotions, and often suffer from label deficiencies, EmoGen lacks semantic coherence and flexibility in practical applications. Unlike EmoGen, the proposed CoEmoGen introduces sentence-level captions as supervision and designs a HiLoRA module in accordance with psychological consensus, resulting in a semantically-coherent and highly scalable EICG pipeline that excels in both semantic clarity and emotional fidelity.

(a) Data construction procedure and examples of filtering      (b) Emotion distribution    (c) Wordcloud

Figure 3: Summary of sentence-level coherent semantic guidance acquisition

# 3 METHODOLOGY

## 3.1 COHERENT SEMANTIC ACQUISITION

EmoSet (Yang et al., 2023) is a recently constructed large-scale visual emotion dataset that follows the popular eight-category Mikels model. It retrieves relevant images from the Internet based on emotions and their synonyms, annotating 118,102 images not only with emotional labels but also with emotion-related visual attributes at different levels, including low-level (*i.e., brightness, colorfulness*), mid-level (*i.e., scene type, object class*), and high-level (*i.e., facial expressions, human actions*). The annotation process is semi-automated; for example, scene types are annotated using a scene recognition model trained on Places365 dataset (Zhou et al., 2017), while object classes are annotated using an object detection model trained on OpenImagesV4 dataset (Kuznetsova et al., 2020). Despite manual intervention for corrections, these word-level labels still suffer from a lack of coherent associations, emotional ambiguity, and missing elements. As a result, EmoGen (Yang et al., 2024), which relies on attribute labels such as scenes or objects for semantic guidance, faces challenges in terms of semantic coherence and the scalability of training corpus.

Recognizing the shortcomings of attribute label supervision, we seek to introduce context-rich and coherent sentence-level captions as guidance. Thanks to the remarkable capabilities demonstrated by recent MLLMs (Li et al., 2025c;b), generating a tailored caption for an image has become cost-effective and feasible. Based on this, we meticulously design a prompt that takes the given image's corresponding emotion label as prior knowledge while enforcing a focus on different levels of emotion-related visual attributes, aligning with those mentioned in EmoSet, thereby encouraging MLLMs to produce a concise yet emotion-rich caption. The designed prompt is as follows:

> `<Image>` This image evokes a strong emotion of `<emotion>`. Provide a *one-sentence caption* that vividly describes the visual details, focusing on elements like *brightness, colorfulness, scene type, object classes, facial expressions, and human actions* that effectively convey and express this emotion.

We investigate the impact of captions obtained from different prompts on subsequent emotional image generation, with details provided in Section 4.3. We equip all images in EmoSet with corresponding captions. However, it is worth noting that due to the inherent hallucinations of MLLMs, which refer to a phenomenon where models generate unrealistic or fabricated content (Bai et al., 2024; Xing et al., 2025), some inaccurate captions are inevitably produced. To address this, we compute the similarity of the preliminarily obtained image-caption pairs in the CLIP space (Radford et al., 2021), discarding the bottom 20% of samples in each category based on similarity rankings to ensure sufficiently reliable semantic guidance. The standardized data construction procedure sets the stage for high scalability, and this entire process, along with examples of filtering, is shown in Figure 3 (a). The emotional distribution of the final training corpus is presented in Figure 3 (b), while a word cloud for each emotion, based on corresponding captions, is shown in Figure 3 (c), with clearer ones provided in Appendix.

## 3.2 COEMOGEN

Based on the aforementioned constructed dataset $\mathcal{D}$, we propose CoEmoGen, an EICG pipeline developed for semantic coherence and high scalability.

**Overview.** Given a sample $\mathcal{D}_i = \{I_i, y_i, c_i\}$, where $I_i$ represents the $i$-th image, $y_i$ denotes the corresponding emotion label, and $c_i$ indicates the generated caption, we first apply one-hot encoding to $y_i$, setting the position of the present category to 1 and all others to 0, obtaining $y_i^o$. The input $y_i^o$

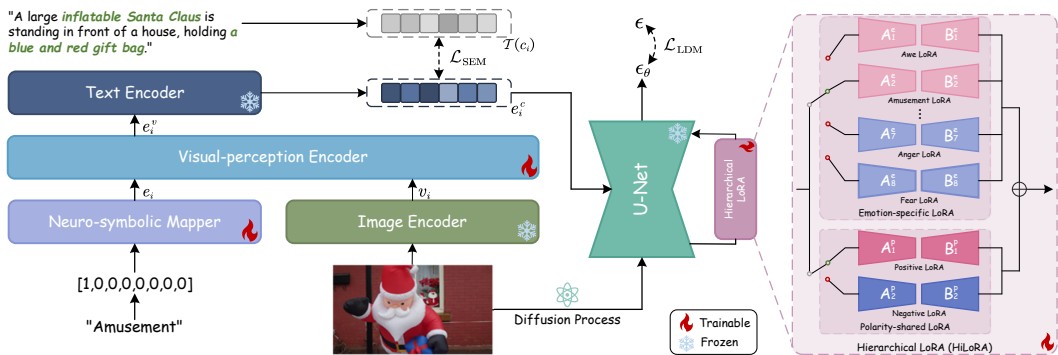

Figure 4: The overall architecture of the proposed CoEmoGen. The one-hot vectors derived from emotion categories first undergo neuro-symbolic mapping and interact with the visual embedding, then are encoded by the text encoder to obtain emotion conditions, which further guide the denoising network U-Net equipped with HiLoRA, which consists of emotion-specific LoRAs and polarity-shared LoRAs.

is then processed by a Neuro-symbolic Mapper composed of fully connected layers with non-linear activations to obtain the emotional descriptor $e_i$. This neuro-symbolic mapping provides greater flexibility and diversity in emotional image generation, which is further discussed in Section 4.3. To achieve more precise alignment with contextually coherent and semantically rich captions, enhancing perception and interaction with visual information is beneficial. Specifically, we fuse the neuro-symbolic vector $e_i$ with the visual embedding $v_i$, which is obtained by encoding $I_i$ through the CLIP image encoder, utilizing a Visual-Perception Encoder primarily based on a cross-attention mechanism. This process can be formulated as follows:

$$e_i^v = \text{Softmax}(\frac{e_i W_q (v_i W_k)^T}{\sqrt{d_0}}) v_i W_v, \tag{1}$$

where $e_i^v$ is visually-enhanced emotion descriptor, $W_q$, $W_k$, and $W_v$ are learnable weights, and $d_0$ is feature dimension. Subsequently, $e_i^v$ is fed into CLIP text encoder to obtain the emotion condition $e_i^c$, which further contributes to the U-Net equipped with Hierarchical LoRA. The entire process described above is illustrated on the left side of Figure 4.

**Hierarchical LoRA.** Low-Rank Adaptation (LoRA) (Hu et al., 2022) is a popular parameter-efficient fine-tuning technique that trains only a small number of parameters through low-rank matrix decomposition, enabling the fine-tuning of pre-trained model weights at a lower computational cost. Its core concept can be formulated as follows:

$$W' = W + \Delta W = W + A \cdot B, \Delta W = A \cdot B \tag{2}$$

where $W$ and $W' \in \mathbb{R}^{d \times d'}$ represent the weights before and after adaptation, respectively, with $A \in \mathbb{R}^{d \times r}$ and $B \in \mathbb{R}^{r \times d'}$ being the introduced low-rank adaptation matrices, satisfying the condition $r \ll \min(d, k)$.

However, in performing EICG, when dealing with the complex concept of representing emotions, a single LoRA proves insufficient. Inspired by psychological observations that emotions with the same polarity share common characteristics in low-level features such as brightness while differing in high-level semantic elements (as shown in Figure 2), we design a hierarchical LoRA (HiLoRA) module, which consists of eight emotion-specific LoRAs $\{\Delta W_j^e\}_{j=1}^8$ tailored to each emotion and two polarity-shared LoRAs $\{\Delta W_k^p\}_{k=1}^2$. These LoRAs function with distinct roles, with only the LoRAs corresponding to the input image's emotion label and its associated polarity activated during updates. Taking *amusement* ($j = 2$) under the *positive* polarity ($k = 1$) as an example, Equation 2 transforms into:

$$W' = W + \Delta W_1^p + \Delta W_2^e = W + A_1^p \cdot B_1^p + A_2^e \cdot B_2^e. \tag{3}$$

Here, $\Delta W_1^p$ is responsible for modeling the high brightness and colorfulness associated with the *positive* polarity, while $\Delta W_2^e$ captures the high-level fine-grained semantics related to *amusement*. The detailed structure of HiLoRA is shown on the right side of Figure 4.

**Loss function.** During training, we optimize the Neuro-symbolic Mapper, Visual-perception Encoder, and HiLoRA while keeping the image encoder, text encoder, and U-Net frozen. Our loss constraints consist of two components. One is the Latent Diffusion Model (LDM) loss, which guides the learning of pixel-level representations:

$$\mathcal{L}_{\text{LDM}} = \mathbb{E}_{\mathcal{E}(\cdot), I_i, e_i^c, \epsilon, t} \left[ \| \epsilon - \epsilon_\theta(z_t, t, \mathcal{E}(I_i), e_i^c) \|_2^2 \right], \tag{4}$$

where $\mathcal{E}(\cdot)$ denotes the latent encoder, $\epsilon_\theta(\cdot)$ indicates the denoising network U-Net, $\epsilon$ refers to the added noise, and $z_t$ is the latent noise at time step $t$. However, applying $\mathcal{L}_{\text{LDM}}$ alone may lead to an excessive focus on pixel-level commonalities and even collapse into specific semantics. To maintain the semantic diversity and coherence of the same emotion, we explicitly approximate the cosine similarity between the emotion condition $e_i^c$ and the encoded sentence-level caption in the CLIP space, introducing the semantic loss $\mathcal{L}_{\text{SEM}}$, formulated as follows:

$$\mathcal{L}_{\text{SEM}} = 1 - \frac{e_i^c \cdot \mathcal{T}(c_i)}{\|e_i^c\| \|\mathcal{T}(c_i)\|}, \tag{5}$$

where $\mathcal{T}$ is the CLIP text encoder. The combination of the above two losses achieves synergy between pixel-level guidance and sentence-level coherent semantic guidance.

During inference, similar to Yang et al. (2024), we obtain visual embedding $v_i$ by randomly sampling from corresponding emotion cluster, which is pre-constructed and represented by a Gaussian distribution, achieving a high degree of diversity.

## 4 EXPERIMENTS

### 4.1 SETTINGS

**Implementation details.** We initialize the latent encoder and denoising network with pre-trained weights from Stable Diffusion v1.5 (Rombach et al., 2022) and choose the pre-trained CLIP ViT-L/14 (Radford et al., 2021) model for both the image and text encoders to ensure a fair comparison. Regarding the model configuration, the hidden layer size of the Neuro-symbolic Mapper is set to 512, the embedding dimension $d_0$ in the Visual-perception Encoder is set to 768, and the rank $r$ in HiLoRA is set to 4. During training, we set the batch size to 1, utilize the AdamW optimizer with $\beta_1 = 0.9$, $\beta_2 = 0.999$, a weight decay of $1e^{-2}$, and a constant learning rate of $1e^{-3}$ for 130,000 iterations. Following Yang et al. (2024), we also adopt a random oversampling strategy to help mitigate class imbalance. All experiments are conducted on two NVIDIA RTX 4090 GPUs with PyTorch. **Metrics.** To comprehensively evaluate model's performance in executing the EICG task in terms of emotion fidelity, semantic clarity, and semantic diversity, following Yang et al. (2024), we generate 1,000 images for each emotion and employ five evaluation metrics: (1) **FID** (Heusel et al., 2017) to measure the distribution distance between generated and real images to assess fidelity, (2) **LPIPS** (Zhang et al., 2018) to assess the overall diversity of the generated images, (3) **Emo-A** (Emotion Accuracy) based on emotion classification accuracy to evaluate the consistency between the generated images and the target emotion, (4) **Sem-C** (Semantic Clarity) to measure the explicitness of the generated image contents, and (5) **Sem-D** (Semantic Diversity) to quantify the richness of content corresponding to each emotion in the generated images. The implementation details of these metrics are in the Appendix.

### 4.2 COMPARISONS

We compare the proposed CoEmoGen with the most relevant and state-of-the-art generation models, including the general image generation model Stable Diffusion, the customized image generation models Textual Inversion and Dreambooth, as well as EmoGen, which is specifically designed for EICG. The fine-tuning of all the above comparison methods is consistent with that in EmoGen. **Qualitative analysis.** Figure 5 presents a qualitative comparison between our proposed CoEmoGen and other methods, with analysis focused on three emotional categories: *awe, anger,* and *contentment* (complete results for all emotions are provided in Appendix). It can be observed that both general and customized image generation models struggle to capture the complex and diverse semantic elements of a specific emotion, often collapsing into a single feature point or exhibiting severe semantic distortion, which makes them highly uncontrollable. In comparison, EmoGen

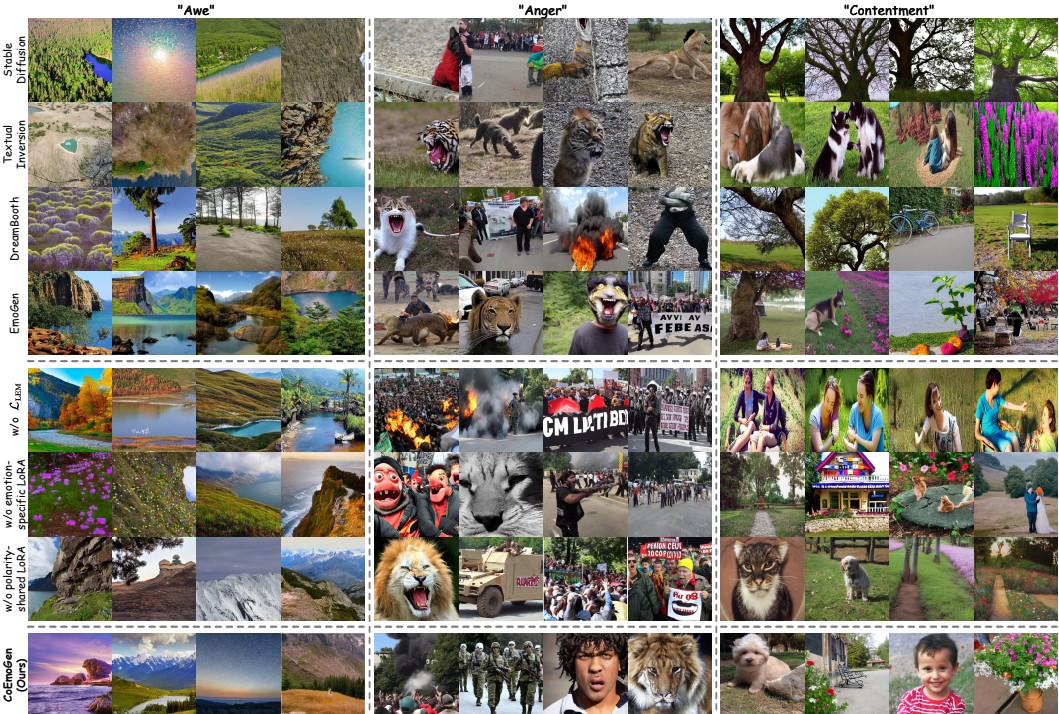

Figure 5: Qualitative comparisons of the proposed CoEmoGen with state-of-the-art generation methods and ablation variants.

| Method | FID↓ | LPIPS↑ | Emo-A↑ | Sem-C↑ | Sem-D↑ |
|---|---|---|---|---|---|
| Stable Diffusion (Rombach et al., 2022) | 44.05 | 0.687 | 70.77% | 0.608 | 0.0199 |
| Textual Inversion (Gal et al., 2022) | 50.51 | 0.702 | 74.87% | 0.605 | 0.0282 |
| DreamBooth (Ruiz et al., 2023) | 46.89 | 0.661 | 70.50% | 0.614 | 0.0178 |
| EmoGen (Yang et al., 2024) | 41.60 | 0.717 | 76.25% | 0.633 | 0.0335 |
| **CoEmoGen (Ours)** | **40.66** | **0.732** | **80.15%** | **0.641** | **0.0349** |
| w/o $\mathcal{L}_{SEM}$ | 50.32 | 0.698 | 65.90% | 0.562 | 0.0255 |
| w/o emotion-specific LoRAs | 45.30 | 0.713 | 75.37% | 0.625 | 0.0308 |
| w/o polarity-shared LoRAs | 41.47 | 0.724 | 78.83% | 0.638 | 0.0336 |

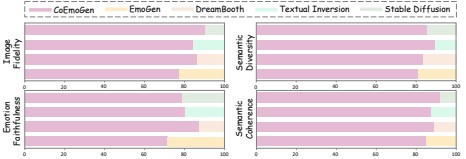

Table 1: Quantitative comparisons with state-of-the-art methods and ablation variants on the EICG task, covering five metrics.

Figure 6: User study results on preference selection between CoEmoGen and a comparative method.

introduces attribute labels as semantic guidance, improving diversity and emotional conveyance, whereas, constrained by the word-level supervision, it unintentionally binds emotions to specific objects or scenes, leading to unnatural collage-like combinations that challenge semantic coherence. For example, *anger* is frequently associated with *fire*, *tigers*, and *wide-open mouths*, and although EmoGen indeed successfully captures these elements, it fails to consider their semantic relationships, resulting in unrealistic outputs such as a *flaming tiger*. In contrast, the proposed CoEmoGen achieves more natural and vivid emotion conveyance, owing to context-rich sentence-level captions that constrain logical connections between elements as guidance, which ensures semantic coherence. Moreover, CoEmoGen preserves diversity while generating images with richer colors and smoother visuals, drawing from the cooperation of emotion-specific LoRAs for fine-grained semantics capture and polarity-shared LoRAs for low-level feature modeling in HiLoRA. Another noteworthy aspect is that CoEmoGen demonstrates versatile photographic composition, rooted in camera-related words like "close-up" in the captions, giving the generated images a stronger sense of perspective and realism. **Quantitative comparison.** Table 1 quantitatively compares CoEmoGen with other methods, demonstrating its overall performance advantage by leading across all five evaluation metrics. Specifically, Emo-A shows an improvement of at least 3.9%, confirming the accuracy of emotional conveyance in the generated images. FID indicates that the generated images better align with the real data distribution, while Sem-C ensures consistent semantic clarity throughout the generation process. Additionally, LPIPS and Sem-D highlight the dual advantages of visual diversity and semantic richness in the generated results. These quantitative findings align with the visualizations in

Figure 5, jointly proving CoEmoGen's superiority in achieving the EICG task. **User study.** To evaluate whether CoEmoGen's generation aligns with human perceptual cognition, we further conduct a comprehensive user study in addition to the aforementioned qualitative and quantitative analyses. The user study involved 17 participants (8 female and 9 male) aged 13-51 with diverse backgrounds. Each participant was presented with two image groups conveying the same target emotions, one generated by CoEmoGen and the other by a comparative method, and asked to select their preference through four criteria-oriented questions: (1) "Which group appears more realistic?" (*Image fidelity*); (2) "Which group better evokes [specific emotion]?" (*Emotion faithfulness*); (3) "Which group shows greater diversity?" (*Semantic diversity*); (4) "Which group demonstrates more natural coherence?" (*Semantic coherence*). As evidenced in Figure 6, CoEmoGen achieves overwhelming preference across all four dimensions, particularly excelling in semantic coherence with an average selection rate of 88.42%. Notably, despite being relatively close to EmoGen in some quantitative metrics, CoEmoGen secures an average user preference rate of 78.67% in direct pairwise comparisons, demonstrating superior alignment with human emotional perception and cognitive intuition, which ultimately translates into stronger emotional resonance in practical applications.

## 4.3 ABLATION STUDY

We conduct ablation studies to explore the effects of each component of the proposed CoEmoGen, as well as the selection of important settings.

We individually remove each component from the complete CoEmoGen and observe the results to explore its effect, which is quantitatively presented in Table 1 and qualitatively shown in Figure 5. **Semantic loss.** It can be observed that when $\mathcal{L}_{\text{SEM}}$ is absent, the model, deprived of the essential semantic guidance and relying solely on $\mathcal{L}_{\text{LDM}}$, excessively focuses on pixel-level reconstruction, even collapsing into specific semantics, leading to significant semantic distortion and a lack of semantic diversity. **Emotion-specific LoRAs.** When emotion-specific LoRAs are removed, the model's ability to capture emotion-specific fine-grained semantics diminishes. Meanwhile, the effect of polarity-shared LoRAs introduces semantic entanglement among emotions belonging to the same polarity. As a result, although the model performs reasonably well in terms of semantic clarity and diversity, the semantic confusion among emotions of the same polarity significantly reduces the accuracy of conveying the target emotion and amplifies the divergence from the true distribution. **Polarity-shared LoRAs.** In comparison, when polarity-shared LoRAs are removed, the quantitative results appear to show only a slight decline compared to the complete CoEmoGen. However, qualitative comparisons reveal that the color saturation and brightness tend to become overly monotonous or unnatural, which further confirms the important role of polarity-shared LoRAs in modeling the low-level features of polarity. In summary, each component of CoEmoGen performs its specific function, and when working in unison, they exhibit the optimal performance in terms of emotional fidelity, semantic diversity and coherence.

Table 2 compares the results of different choices of important settings. **Type of caption.** To obtain the most suitable captions as sentence-level semantic guidance, we explore four prompts to drive the MLLMs, controlling whether the captions focus on emotion-related elements and contain complete details, with further details provided in the Appendix. As shown in Table 2 (a), adding emotional priors to the prompt promotes consistent improvement in the model's ability, highlighting the importance of focusing on emotional elements. However, removing the restriction to generate only one sentence and allowing the inclusion of as many details as possible leads to performance degradation, likely due to the noise introduced by irrelevant elements and the increased risk of hallucinations in MLLMs with longer captions, making model convergence more difficult. Therefore, we ultimately choose a prompt with emotional priors and the constraint of generating a single sentence. **Type of input.** We compare three different types of input representa-

| Emotional Prior | Details | FID↓ | LPIPS↑ | Emo-A↑ | Sem-C↑ | Sem-D↑ |
|---|---|---|---|---|---|---|
| ✗ | ✗ | 43.32 | 0.714 | 75.29% | 0.612 | 0.0303 |
| ✓ | ✗ | **40.66** | **0.732** | **80.15%** | **0.641** | **0.0349** |
| ✗ | ✓ | 44.91 | 0.716 | 73.68% | 0.604 | 0.0315 |
| ✓ | ✓ | 42.58 | 0.724 | 77.42% | 0.628 | 0.0338 |

(a) Type of captions for semantic guidance.

| Type of Input | FID↓ | LPIPS↑ | Emo-A↑ | Sem-C↑ | Sem-D↑ |
|---|---|---|---|---|---|
| Learnable | 43.77 | 0.705 | 78.49% | 0.593 | 0.0289 |
| Text | 42.38 | 0.697 | 78.92% | 0.611 | 0.0275 |
| One-hot | **40.66** | **0.732** | **80.15%** | **0.641** | **0.0349** |

(b) Type of inputs for emotion representation.

| Type of $\mathcal{L}_{\text{SEM}}$ | FID↓ | LPIPS↑ | Emo-A↑ | Sem-C↑ | Sem-D↑ |
|---|---|---|---|---|---|
| MAE | 42.40 | 0.721 | 78.83% | 0.626 | 0.0311 |
| MSE | 41.87 | 0.725 | 79.04% | 0.618 | 0.0327 |
| K-L | 48.36 | 0.693 | 70.01% | 0.609 | 0.0272 |
| Cosine | **40.66** | **0.732** | **80.15%** | **0.641** | **0.0349** |

(c) Type of semantic constraint mechanisms in $\mathcal{L}_{\text{SEM}}$.

Table 2: Quantitative results of CoEmoGen with different choices of important settings. Default settings are marked in gray.

tions, including learnable class vectors, frozen text encodings of emotion class names, and the one-hot vectors with the Neuro-symbolic Mapper. As shown in Table 2 (b), experimental results reveal significant limitations in the first two types in terms of diversity, which is due to the representation bottleneck from the limited capacity of learnable vectors and over-constraining semantics in the embedding space from the frozen text encoder. In contrast, the Neuro-symbolic Mapper expands the latent space exploration while maintaining semantic consistency through a differentiable symbolic reasoning mechanism, enabling greater diversity. **Design of $\mathcal{L}_{\text{SEM}}$.** In the design of the semantic loss $\mathcal{L}_{\text{SEM}}$, constructing an effective semantic constraint mechanism is a question worth considering. To address this, we systematically compare the performance differences of four constraint methods: Mean Absolute Error (MAE), Mean Squared Error (MSE), Kullback-Leibler Divergence (K-L divergence), and Cosine Similarity. As shown in Table 2 (c), Cosine Similarity achieves the best performance, owing to its inherent advantage in measuring semantic similarity in high-dimensional vector spaces, enabling more precise guidance for the model to capture deep semantic relationships. Therefore, we choose it by default.

### 4.4 Scalability of CoEmoGen

Thanks to MLLMs facilitating the accessibility of sentence-level semantic guidance captions, Co-EmoGen exhibits high scalability compared to EmoGen, which relies on high acquisition costs and limited word-level attribute labels for supervision. This feature enables CoEmoGen to incorporate any emotionally evocative images into the training corpus.

To illustrate this, we construct a large-scale emotional art image dataset, **EmoArt**. Specifically, we first collect approximately 100,000 artistic images from WikiArt, covering 129 artists, 11 genres, and 27 styles. Then, we utilize a classifier pre-trained on EmoSet to predict the emotional categories of these artistic images, setting an emotion confidence threshold of 0.75, which ultimately filters out 13,633 strongly emotion-representative samples. Notably, due to the low-frequency presence of *excitement* and *disgust* in artistic expressions (accounting for less than 1%) (Pearce et al., 2016), EmoArt focuses on the remaining six emotion categories. A visualization of EmoArt's images across different emotions is provided in Appendix. Next, following the same process as Figure 3 (a), we obtain

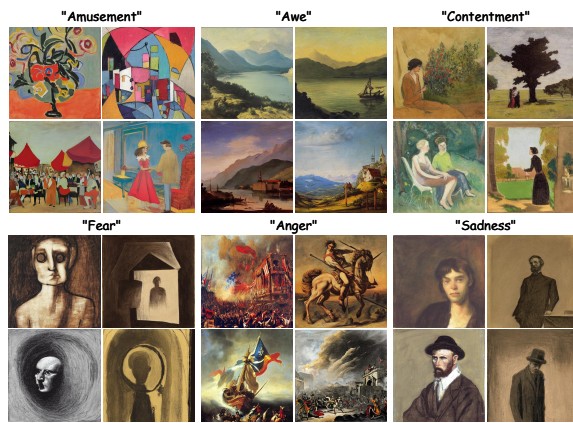

Figure 7: Visualization of emotionally evocative artistic images generated by EmoArt-driven CoEmoGen for each emotion.

reliable sentence-level captions for each image, which are then utilized to train CoEmoGen. Figure 7 presents the results generated by EmoArt-driven CoEmoGen, demonstrating its ability to create emotionally faithful and stylistically diverse artistic images for each emotion category, offering endless inspiration for artists in crafting emotionally evocative artworks.

The construction of EmoArt intuitively showcases CoEmoGen's high flexibility and scalability, allowing users to follow a standardized construction paradigm to effortlessly customize diverse emotion-rich datasets and generate emotionally stimulating images tailored to their specific needs.

### 4.5 Applications

Given its strong emotional content generation capability, we further explore CoEmoGen's intriguing applications. **Emotion Transfer.** By integrating the emotional representations learned by Co-EmoGen with common neutral elements, we achieve targeted emotion transfer, as shown in Figure 8. Excitingly, this results in a seamless fusion, where the emotional semantics are incorporated while staying true to the original neutral elements. Importantly, instead of rigidly overlaying emotion-related elements onto the neutral ones, this process perceives the semantic characteristics of the neutral elements and adaptively adjusts their textures, colors, and layouts to ensure overall coher-

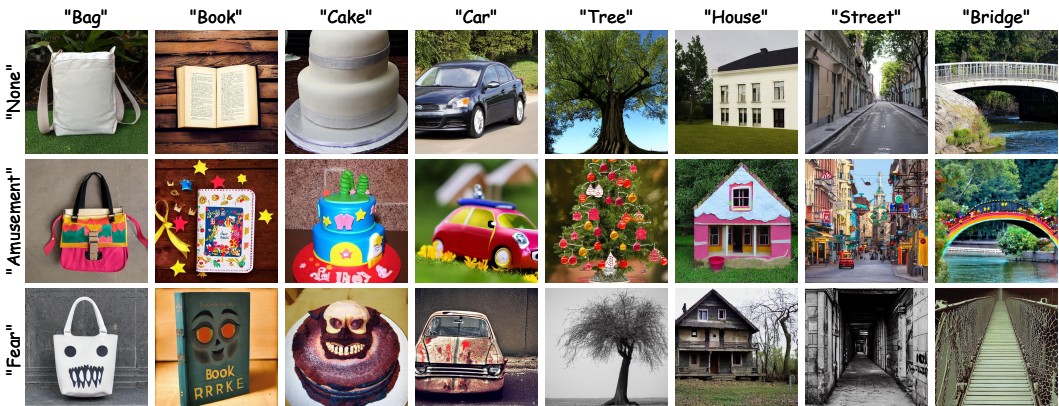

Figure 8: Visualization of emotion transfer by fusing emotion representations with neutral elements.

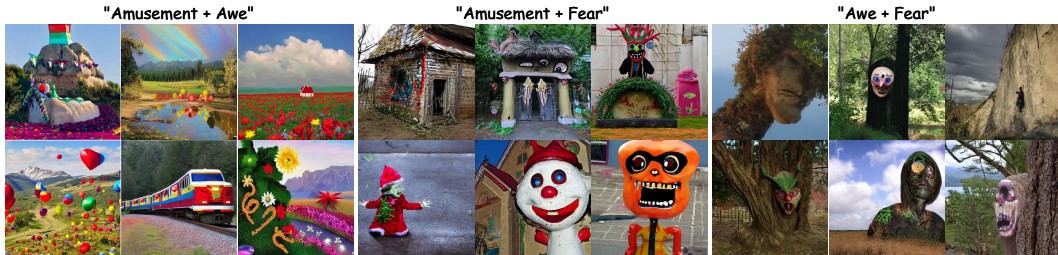

Figure 9: Visualization of emotion fusion by combining different emotion representations.

ence. For example, in the fear transfer applied to a street scene, CoEmoGen enhances shadow contrast and spatial depth, generating a dark and deep corridor that aligns with real-world lighting principles rather than mechanically adding horror symbols, ensuring a gradual and immersive infusion of emotional semantics and offering promising potential for interpretable image emotional editing. **Emotion Fusion.** We also explore the possibility of combining different emotional representations, as shown in Figure 9, which illustrates the pairwise combinations of *amusement, awe,* and *fear*, showcasing a visually rich narrative intertwined with complex emotions. For example, in *amusement+awe*, colorful balloons and a rainbow in mountain valley create a fantastical yet dignified visual language, while in *awe+fear*, the transformation of mountain contours shapes the rock formations into a devil-like face, collectively reinforcing a sense of wonder and fear. When fusing emotions, we are essentially organically integrating emotional semantics at visual level, allowing CoEmoGen not only to generate single-emotion images but also to explore more multi-dimensional emotional experiences.

## 5 CONCLUSION AND FUTURE WORK

In this paper, we develop CoEmoGen, a novel EICG pipeline excelling in semantic coherence and high scalability. Leveraging MLLMs, we obtain reliable context-rich sentence-level captions for the images in EmoSet to serve as semantically-coherent guidance. Referring to psychological theory, we design a HiLoRA module, including polarity-shared LoRAs to model common low-level features and emotion-specific LoRAs to capture high-level exclusive semantics. We demonstrate the superiority of the proposed CoEmoGen in EICG from qualitative, quantitative, and user study perspectives, and further collect and construct a large-scale emotional art image dataset, EmoArt, to concretely showcase high scalability. In future work, we plan to investigate the denoising process in depth, explicitly analyzing the dynamic shifts of attention toward emotion-related attributes across different denoising stages.

## ACKNOWLEDGMENTS

This work was supported by the Guangzhou-HKUST(GZ) Joint Funding Program (Grant No. 2023A03J0008), Education Bureau of Guangzhou Municipality. It was also supported by Jiangsu Industrial Technology Research Institute (JITRI) and Wuxi National High-Tech District (WND).

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

# APPENDIX

## A   ADDITIONAL VISUALIZATIONS

We present qualitative comparisons of the proposed CoEmoGen with state-of-the-art generation methods and ablation variants across all emotion categories in Figure 10 to 13, from which conclusions aligned with those discussed in Section 4.2 of the main manuscript can be drawn, demonstrating the consistency. To better illustrate the newly collected large-scale emotional art image dataset, EmoArt, we also provide stylistically diverse sample examples for each emotion category in Figure 14. Besides, we generate word clouds for each emotion category based on the corresponding captions, allowing us to observe the high-frequency words or phrases in each category, as shown in Figure 15.

## B   PROMPTS

To obtain the most suitable and effective sentence-level captions as coherent semantic guidance, we conduct an ablation study in Section 4.3 of the main manuscript, exploring the impact of captions derived from different versions of prompts. The key differences in these prompts lie in two aspects: whether they include emotional priors and whether they restrict the generation to a single sentence. The specific details of the adopted prompts are provided in Table 3. Additionally, considering the differences in emotional representation elements between artistic and natural images, we design a customized prompt specifically for generating captions for artworks in EmoArt, which guides MLLMs to focus on key elements that convey emotions in artistic images, as detailed in Table 4.

---

- "`<Image>` Provide a one-sentence caption that vividly describes the visual details of this image."
- "`<Image>` This image evokes a strong emotion of `<emotion>`. Provide a one-sentence caption that vividly describes the visual details, focusing on elements like brightness, colorfulness, scene type, object classes, facial expressions, and human actions that effectively convey and express this emotion."
- "`<Image>` Write a terse but informative summary of the picture."
- "`<Image>` This image evokes a strong emotion of `<emotion>`. Provide a terse but informative summary that vividly describes the visual details, focusing on elements like brightness, colorfulness, scene type, object classes, facial expressions, and human actions that effectively convey and express this emotion."

---

Table 3: Different versions of prompts used to drive MLLMs for generating semantically coherent sentence-level captions.

---

`<Image>` This *artwork* evokes a strong emotion of `<emotion>`. Provide a *one-sentence caption* that vividly describes the visual details, focusing on elements like *artistic style, material textures, composition balance, symbolic elements, dominant color choices, expressive brushstrokes, and dramatic light/shadow contrasts* that effectively convey and express this emotion.

---

Table 4: A customized prompt specifically designed for generating captions for artworks in EmoArt.

## C   EVALUATION METRICS

To comprehensively evaluate the model's performance in executing the EICG task, we employ five evaluation metrics, which are introduced below.

**FID (Fréchet Inception Distance) (Heusel et al., 2017).** FID is a metric employed to evaluate the performance of generative models by measuring the distribution similarity between generated and real images in feature space, which can be formulated as:

$$\text{FID} = \|\mu_r - \mu_g\|^2 + \text{Tr}\left(\Sigma_r + \Sigma_g - 2(\Sigma_r \Sigma_g)^{1/2}\right), \tag{6}$$

where $\mu_r$ and $\mu_g$ represent the means of features extracted from real and generated images using the Inception network (Szegedy et al., 2015), respectively, while $\Sigma_r$ and $\Sigma_g$ denote the corresponding covariance matrices. A lower FID value indicates a closer match between the statistical properties of the generated and real images. It is important to note that due to the inherent limitations of FID (Jayasumana et al., 2024), we need to incorporate other metrics for a more accurate evaluation.

**LPIPS (Learned Perceptual Image Patch Similarity) (Zhang et al., 2018).** We employ LPIPS, a perceptual metric aligning with human vision by measuring image differences through deep feature correlations, to evaluate the overall diversity of generated images. Specifically, we randomly select $P_i$ pairs of generated images for each emotion category and calculate the LPIPS score by averaging the perceptual distances between these pairs. Finally, the overall LPIPS metric is obtained by averaging the LPIPS scores across all emotion categories, as formalized below:

$$\text{LPIPS} = \frac{1}{C}\frac{1}{P_i}\sum_{i=1}^{C}\sum_{p=1}^{P_i}\text{LPIPS}(a_p^i, b_p^i), \tag{7}$$

where $C$ denotes the total number of emotion categories, $P_i$ represents the number of sampled image pairs for the $i$-th emotion, and $\text{LPIPS}(a_p^i, b_p^i)$ indicates the perceptual similarity score between the $p$-th pair of images $a$ and $b$ the $i$-th emotion category.

**Emo-A (Emotion Accuracy).** Emo-A is introduced as a metric to quantitatively evaluate emotion faithfulness. It leverages a pre-trained emotion classifier to predict the emotion category of generated images, which are then compared against the target emotion. The final accuracy is computed exclusively based on samples with correct emotion alignment.

**Sem-C (Semantic Clarity).** To quantify semantic clarity, Sem-C employs a pre-trained object classifier (He et al., 2016) from ImageNet (Deng et al., 2009) and a pre-trained scene classifier (He et al., 2016) from Places365 (Zhou et al., 2017) to classify the generated images. The final metric is computed by taking the highest probability between these two classifiers, as formalized below:

$$\text{Sem-C} = \frac{1}{N}\sum_{n=1}^{N}\max\left(v_{\text{object}}(x_n), v_{\text{scene}}(x_n)\right) \tag{8}$$

where $N$ represents the total number of generated images, where $v_{object}$ and $v_{scene}$ denote the object and scene classifiers, respectively.

**Sem-D (Semantic Diversity).** To measure semantic diversity, we randomly sample $P_i$ pairs of generated images for each emotion category and compute the Mean Squared Error (MSE) between their CLIP image embeddings for each pair, averaging the results to obtain the Sem-D score for that emotion. The overall Sem-D metric is then obtained by averaging these scores across all emotion categories, as formalized below:

$$\text{Sem-D} = \frac{1}{C}\frac{1}{P_i}\sum_{i=1}^{C}\sum_{p=1}^{P_i}\text{MSE}\left(\text{CLIP}_I(a_p^i), \text{CLIP}_I(b_p^i)\right), \tag{9}$$

where $C$ denotes the total number of emotion categories, $P_i$ represents the number of sampled image pairs for the $i$-th emotion, and $\text{CLIP}_I$ indicates the CLIP image encoder.

## D    THE USE OF LLMs

In this work, LLMs were used to assist with grammar checking and sentence refinement to improve the clarity and overall readability of the manuscript.

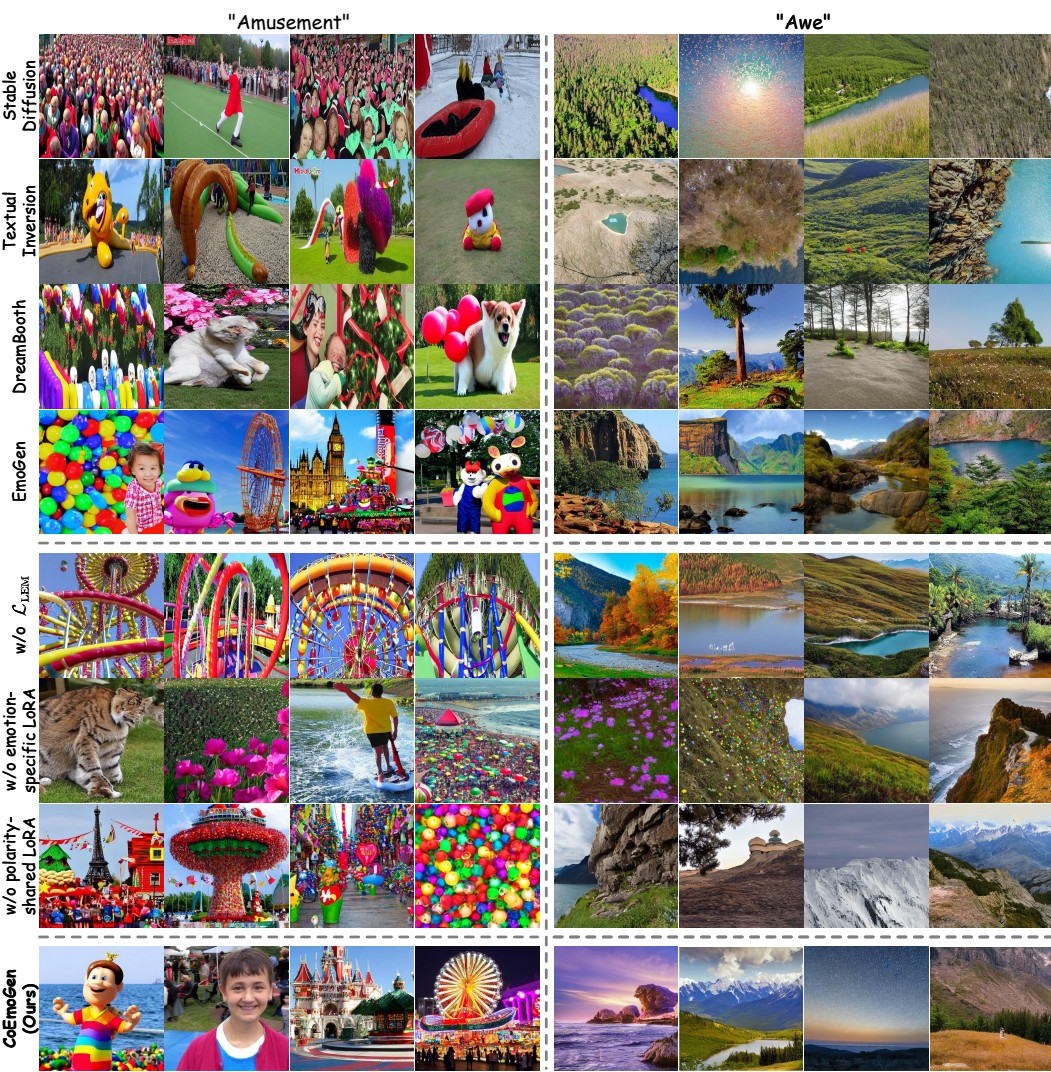

Figure 10: Qualitative comparisons of the proposed CoEmoGen with state-of-the-art generation methods and ablation variants on *amusement* and *awe* emotions.

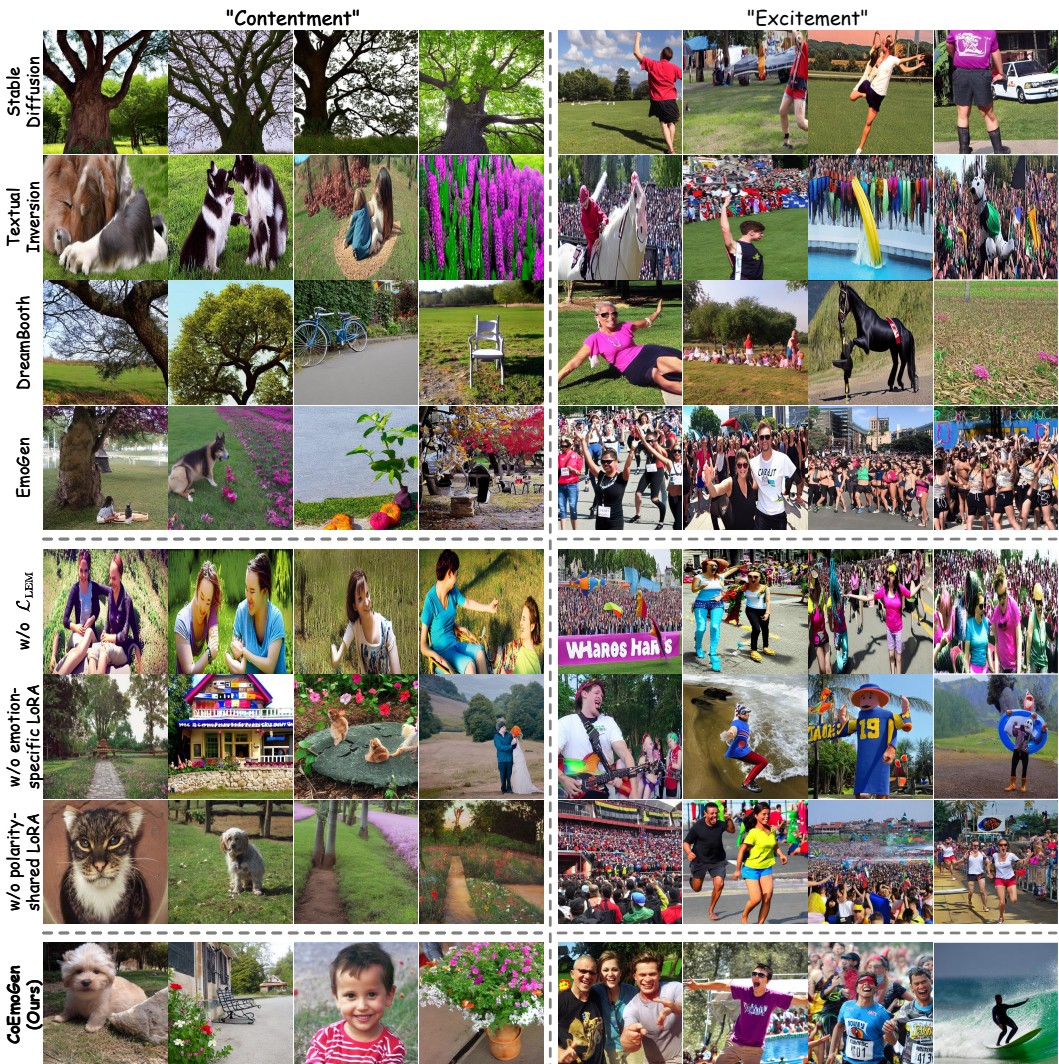

Figure 11: Qualitative comparisons of the proposed CoEmoGen with state-of-the-art generation methods and ablation variants on *contentment* and *excitement* emotions.

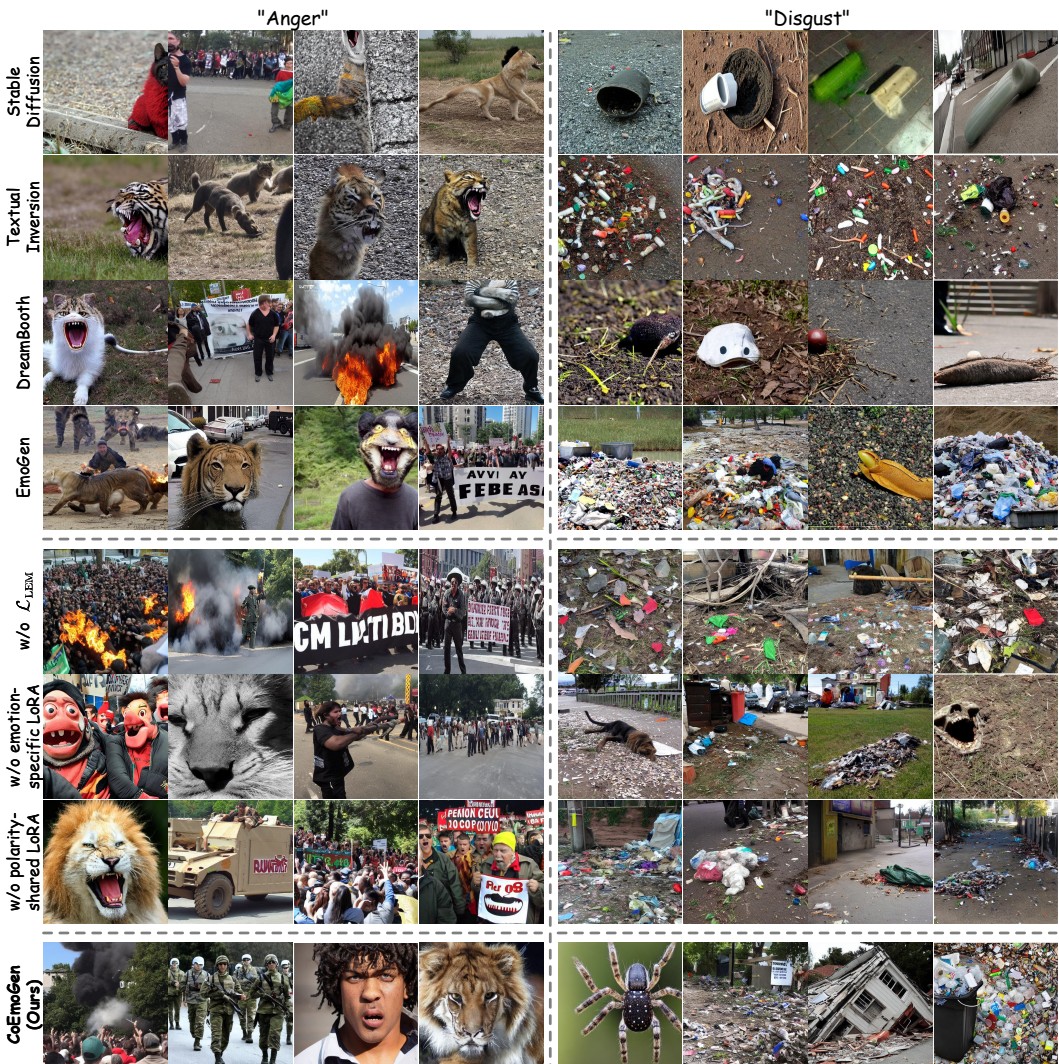

Figure 12: Qualitative comparisons of the proposed CoEmoGen with state-of-the-art generation methods and ablation variants on *anger* and *disgust* emotions.

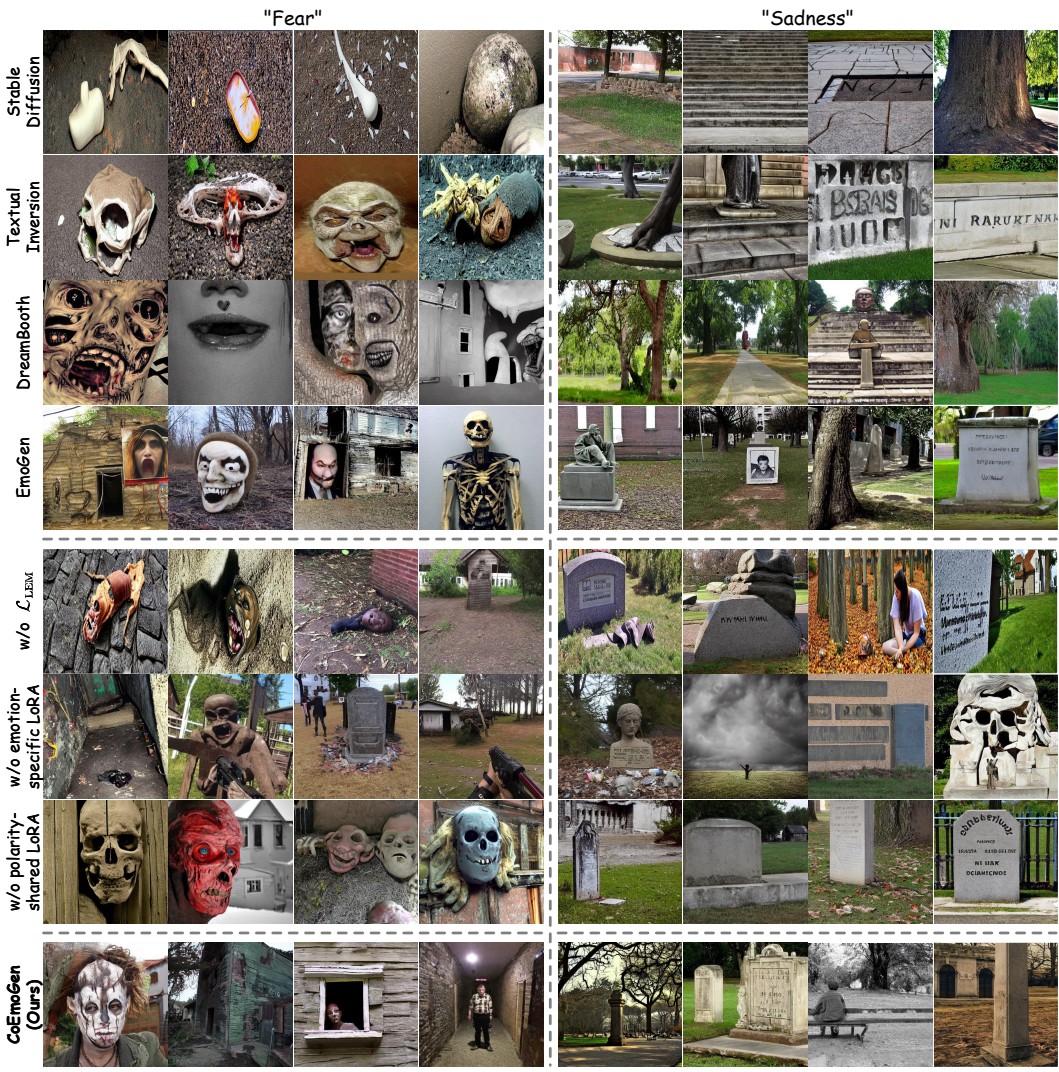

Figure 13: Qualitative comparisons of the proposed CoEmoGen with state-of-the-art generation methods and ablation variants on *fear* and *sadness* emotions.

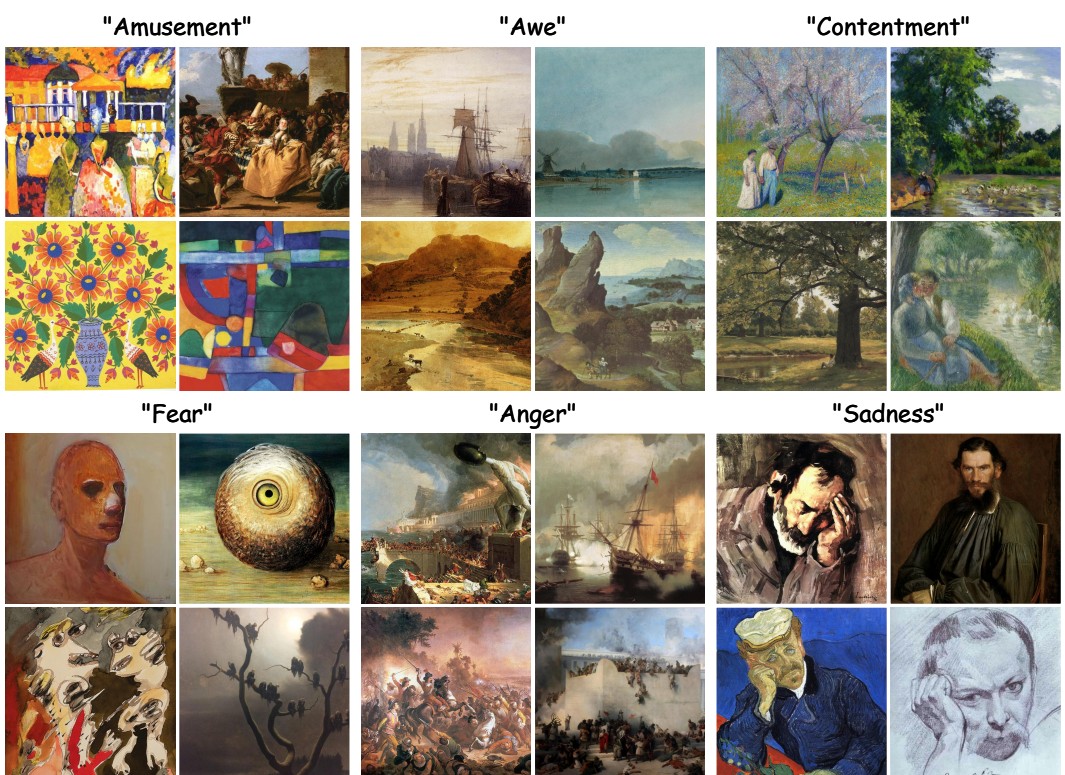

Figure 14: Sample examples of each emotion in the constructed EmoArt.

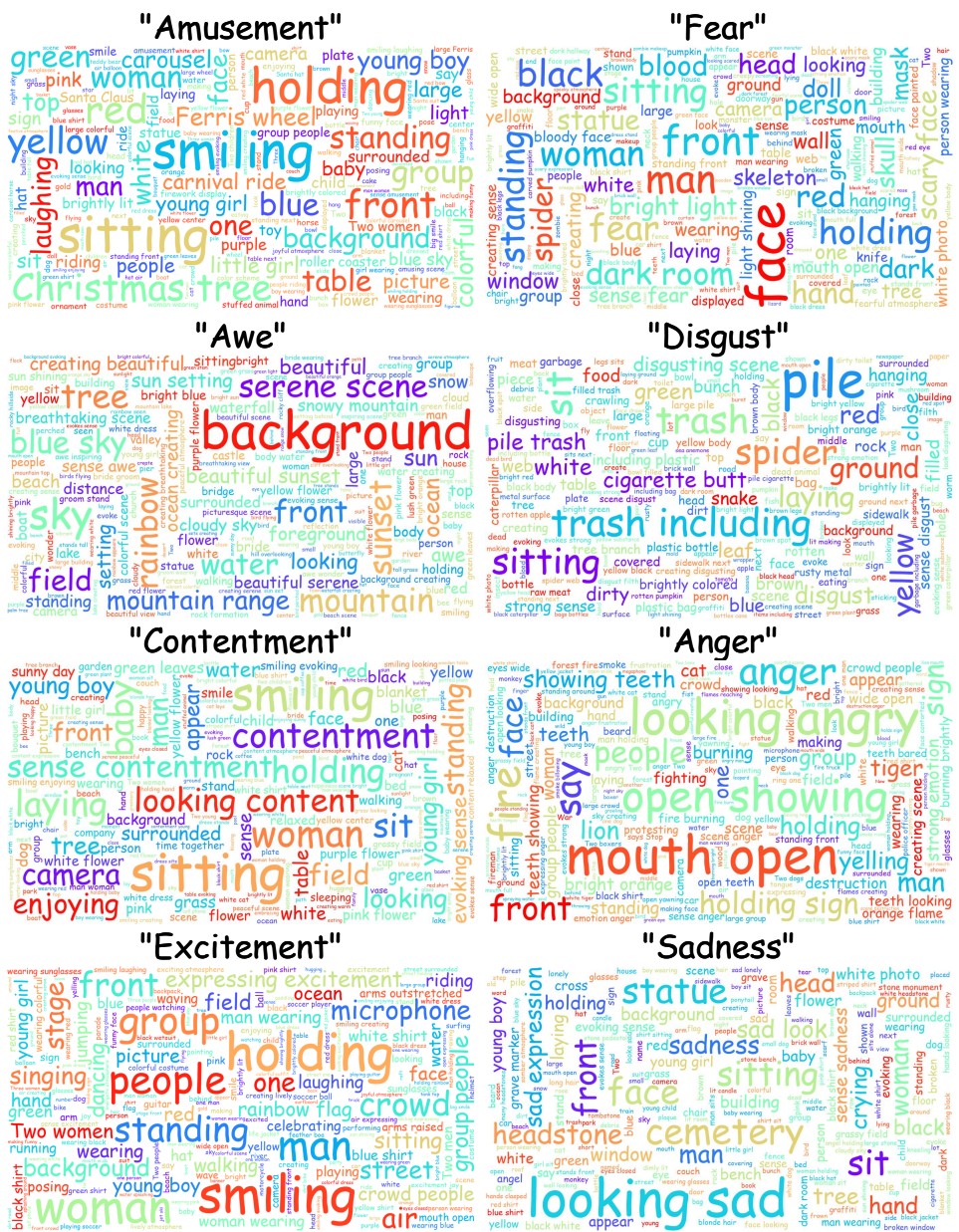

Figure 15: Word clouds generated from captions corresponding to each emotion category.

