# OpenReview forum: "CoEmoGen: Towards Semantically-Coherent and Scalable Emotional Image Content Generation"
_ICLR.cc/2026/Conference — ICLR 2026 Poster_

### Official Review · Reviewer_LzoM · 2025-10-24

**Soundness:** 4
**Presentation:** 3
**Contribution:** 3
**Rating:** 4
**Confidence:** 3

**Summary:**

The paper introduces CoEmoGen, a diffusion-based approach to generate emotion-conditioned images by utilizing hierarchical LORAs.

**Strengths:**

1. Novel approach of using hierarchy for emotions by having two levels, positive/negative for level one and Sad/Happy, etc. for level 2.
2. The results show the superior performance of the proposed approach compared to the baselines.

**Weaknesses:**

1. The final adapted weight is simply a sum of these LoRAs, which means both adapters are applied in parallel rather than sequentially. This design is more of a multi-branch or compositional LoRA, rather than a truly hierarchical one. A hierarchical LoRA would involve one LoRA’s output feeding into another, or a structured hierarchy where adapters are activated conditionally or at different network depths.
2. A simple baseline where the emotion label is fed as input to an LLM, which generates a caption for that emotion, and then is fed to a T2I model is missing. I tried this with Flux and the results were reasonable. The results from this experiment would confirm if the problem requires a specific training approach, thereby strengthening the paper's claim.
3. Line 425: "we construct the first large-scale emotional art image dataset". This is incorrect, as [1,2] have done it before, and the data is manually verified. Maybe better to use this dataset.

[1] ArtEmis: Affective Language for Visual Art: Panos Achlioptas, Maks Ovsjanikov, Kilichbek Haydarov, Mohamed Elhoseiny, Leonidas Guibas

[2] It is Okay to Not be Okay: Overcoming Emotional Bias in Affective Image Captioning by Contrastive Data Collection: Youssef Mohamed, Faizan Farooq Khan, Kilichbek Haydarov, Mohamed Elhoseiny

**Questions:**

1. How many images were evaluated via the human study?
2. How is 'targeted emotion transfer' in Fig. 8 achieved?

---

> ### Author Response · Authors · 2025-11-28
> **Response to Reviewer LzoM (1/2)**
>
> 1. **About hierarchy (W1) :**
>
>    Thank you for your valuable comments. We would like to clarify that the “hierarchy” in HiLoRA does not refer to the network architecture, but rather to the hierarchical emotional cognition derived from the psychological Mikels model, which is divided into two levels: emotional polarity and discrete emotion categories. Correspondingly, we designed parallel functionally specialized LoRAs in HiLoRA, namely the polarity-shared LoRA and the emotion-specific LoRA, which align with this cognitive hierarchy, respectively for modeling polarity-shared low-level common features and for capturing emotion-specific high-level fine-grained semantics.
>
>    Additionally, we further experimented with a sequential alternative in HiLoRA for ablation analysis, and as shown in the table below, this sequential design leads to performance drops across all metrics compared with the original parallel design. This is because sequential stacking entangles the representations of polarity-shared and emotion-specific features, which leads to interference and confusion; in contrast, parallel fusion aligns more naturally with the way emotional expression is formed, namely by integrating shared common features with fine-grained emotion-specific distinctions. We will include this discussion in the final version for clarification.
>
>    | Setting         | FID ↓ | LPIPS ↑ | Emo-A ↑ | Sem-C ↑ | Sem-D ↑ |
>    | --------------- | ----- | ------- | ------- | ------- | ------- |
>    | Sequential      | 42.85 | 0.719   | 77.04%  | 0.627   | 0.0326  |
>    | Parallel (Ours) | 40.66 | 0.732   | 80.15%  | 0.641   | 0.0349  |
>
> 2. **About an additional baseline (W2) :**
>
>    We appreciate your suggestion and supplement an additional baseline that attempts to perform the EICG task using a combination of an LLM and a T2I model. Specifically, we first use an LLM to generate 1000 captions corresponding to each emotion by prompting: “Please generate a caption for an image that conveys a sense of [emotion],” and then feed these captions into a T2I model to generate the images. We choose Qwen2.5-7B and GPT-5 as the LLMs and Stable Diffusion v1.5 and FLUX.1 as the T2I models, with the results shown in the table below.
>
>    | Method                             | FID ↓ | LPIPS ↑ | Emo-A ↑ | Sem-C ↑ | Sem-D ↑ |
>    | ---------------------------------- | ----- | ------- | ------- | ------- | ------- |
>    | Qwen2.5-7B + Stable Diffusion v1.5 | 52.26 | 0.654   | 64.83%  | 0.591   | 0.0176  |
>    | Qwen2.5-7B + FLUX.1                | 48.03 | 0.665   | 68.29%  | 0.638   | 0.0207  |
>    | GPT-5 + Stable Diffusion v1.5      | 48.87 | 0.679   | 66.59%  | 0.616   | 0.0263  |
>    | GPT-5 + FLUX.1                     | 46.12 | 0.693   | 71.21%  | 0.654   | 0.0298  |
>    | CoEmoGen (Ours)                    | 40.66 | 0.732   | 80.15%  | 0.641   | 0.0349  |
>
>    It can be observed that the generation quality of the LLM→caption→T2I pipeline is closely tied to the capabilities of both the LLM and the T2I model. A stronger LLM can generate more accurate emotion-oriented captions, while a stronger T2I model can produce higher-quality images based on these captions, making the final performance on the EICG task jointly determined by both components.
>
>    However, even the strongest combination, GPT-5 + FLUX.1, still shows a significant gap compared with CoEmoGen. Fundamentally, this is because providing only the emotion label is inadequate for guiding the LLM to produce sufficiently rich captions, which limits image diversity, and off-the-shelf T2I models are also limited in understanding emotion-related semantics in the captions and fail to translate emotional cues into emotionally aligned visual content, weakening the precision of emotional expression in the generated images. Therefore, these findings further confirm the necessity of employing a dedicated method like CoEmoGen to effectively accomplish the EICG task. We will incorporate this discussion in the revised version.

---

> ### Author Response · Authors · 2025-11-28
> **Response to Reviewer LzoM (2/2)**
>
> 3. **About emotional art dataset (W3) :**
>
>    Thank you very much for your reminder. We acknowledge that our previous description of EmoArt was not sufficiently precise, and we will narrow the wording accordingly to clarify its specific focus on the EICG task in the revised version. We will also incorporate a clearer discussion of both ArtEmis [R1] and ArtEmis v2.0 [R2], which indeed represent important and high-quality large-scale emotional art datasets.
>
>    What we would like to emphasize is that the goal of constructing EmoArt is not the dataset itself, but rather to serve as a demonstrative case illustrating the high flexibility and scalability of CoEmoGen, which provides a standardized paradigm that allows users to effortlessly customize diverse emotion-rich datasets and generate emotionally stimulating images tailored to their specific needs. To better align with prior work and further validate CoEmoGen’s scalability, we will additionally explore using the ArtEmis family of datasets as training corpora in the revised version, highlighting its flexibility and adaptability.
>
>    [R1] Achlioptas, Panos, et al. "Artemis: Affective language for visual art." *Proceedings of the IEEE/CVF Conference on Computer Vision and Pattern Recognition*. 2021.
>
>    [R2] Mohamed, Youssef, et al. "It is okay to not be okay: Overcoming emotional bias in affective image captioning by contrastive data collection." Proceedings of the IEEE/CVF conference on computer vision and pattern recognition. 2022.
>
> 4. **About user study (Q1) :**
>
>    During the user study, each participant was presented with two image groups conveying the same target emotions, one generated by CoEmoGen and the other by a comparative method, and asked to select their preference. Of these, each image group contained 16 images, and direct pairwise comparisons were conducted with four comparative methods covering eight target emotions, thus involving a total of 16 × 2 × 4 × 8 = 1,024 images, half generated by CoEmoGen and half by the comparative methods.
>
> 5. **About targeted emotion transfer (Q2) :**
>
>    During targeted emotion transfer, we concatenate the emotion descriptor of the target emotion, obtained from the Visual-perception Encoder, with the embeddings of the given neutral elements and feed this into the text encoder to obtain the condition that contributes to the U-Net, while simultaneously activating the polarity-shared LoRA and emotion-specific LoRA corresponding to the target emotion, with all other processes remaining default. This enables a seamless and coherent fusion, where emotional semantics are incorporated while staying true to the original neutral elements, by perceiving and adaptively adjusting their semantic characteristics, rather than merely mechanically overlaying emotion-related symbols.

---

### Official Review · Reviewer_2zyk · 2025-10-30

**Soundness:** 2
**Presentation:** 2
**Contribution:** 3
**Rating:** 4
**Confidence:** 4

**Summary:**

The paper proposes CoEmoGen, a semantically coherent and scalable framework for emotional image generation. It leverages MLLM-generated sentence-level captions to provide rich emotional supervision and introduces a psychologically inspired HiLoRA architecture that integrates polarity-level and emotion-specific representations. Experiments show consistent improvements across quantitative metrics and human evaluations, confirming enhanced emotional fidelity and semantic coherence.

**Strengths:**

1. The introduction of MLLM-generated sentence-level captions offers an effective and scalable framework for automated emotional annotation.

2. The proposed HiLoRA architecture elegantly integrates shared polarity-level and emotion-specific representations to achieve psychologically grounded emotional modeling.

3. By constructing the EmoArt dataset, the work successfully extends emotional image generation beyond photographic realism into artistic and creative domains.

**Weaknesses:**

1. The MLLM-generated captions may introduce subtle semantic biases or hallucinations, yet there is no human annotation or validation to assess their linguistic accuracy or emotional authenticity.

2. The use of a frozen CLIP text encoder stabilizes training but limits emotional adaptability, as CLIP’s language space is not optimized for affective or psychological semantics.

3. The visual–emotion bias inherent in EmoSet (e.g., fear = terrifying faces) is amplified by multimodal captions and further reinforced through independent emotion-specific LoRA modules. Without textual correction during inference, these biases are directly reflected in the generated results, leading to symbolic and monotonous emotional expressions.

4. Although sentence-level captions from MLLMs aim to enhance semantic coherence, the frequent appearance of neutral phrases (e.g., mouth open) in the word cloud suggests that the model primarily learns low-level visual co-occurrences rather than abstract emotional semantics, which weakens CoEmoGen’s theoretical contribution and generalization ability in genuine affective generation.

5. The figures and tables in the paper use very small fonts, which significantly hinders readability and clarity of presentation.

**Questions:**

1. How do the authors ensure the semantic validity and emotional reliability of the MLLM-generated captions, given that no human verification or annotation was conducted?

2. Why did the authors choose not to include the CLIP text encoder in training, or alternatively, to constrain the Visual-Perception Encoder’s output to align with the text embedding space?

3. Given that EmoSet’s visual–emotion bias (e.g., “fear = terrifying faces”) and the frequent presence of neutral phrases in captions (e.g., “mouth open”) may reinforce low-level correlations, how might the authors mitigate such biases to encourage abstract and contextually rich emotional understanding?

---

> ### Author Response · Authors · 2025-11-28
> **Response to Reviewer 2zyk (1/2)**
>
> 1. **About quality of MLLM-generated captions (W1&Q1) :**
>
>    During coherent semantic acquisition with MLLM, we mitigate inaccuracies caused by inherent hallucinations and ensure semantic reliability by leveraging the similarity of image-caption pairs in the CLIP latent space and applying a threshold for filtering. To further investigate the quality of the generated captions, we randomly sample 50 captions from each of the 8 emotion categories (400 samples in total) and invite three human annotators to individually rate them on linguistic accuracy and emotional authenticity using a five-point Likert scale (1 = poor, 5 = excellent). The statistical results of this evaluation are shown in the table below. It can be observed that over 87% of the samples receive high scores (≥4) in both aspects, while the proportion of low-scoring outlier samples remains below 5%. We also calculate the Intraclass Correlation Coefficient (ICC) among the human annotators, which is 0.93 (closer to 1 indicates better agreement), demonstrating a high level of inter-annotator agreement.
>
>    | Score | Linguistic accuracy | Emotional authenticity |
>    | ----- | ------------------- | ---------------------- |
>    | 1     | 1.42%               | 1.92%                  |
>    | 2     | 2.08%               | 2.67%                  |
>    | 3     | 4.33%               | 8.25%                  |
>    | 4     | 28.08%              | 27.58%                 |
>    | 5     | 64.08%              | 59.58%                 |
>
>    These results demonstrate that high-quality MLLM-generated captions can be obtained through our standardized data construction procedure without human intervention. We infer that this is because captioning itself is a relatively simple and not highly challenging task; as long as the MLLM (we use Qwen2-VL-7B by default) reaches a certain level of capability, the generated captions are reliable and credible. Compared to time-consuming, labor-intensive, and costly human verification or annotation, our automated paradigm represents a more cost-effective solution. We will incorporate this discussion in the revised version.
>
> 2. **About freezing CLIP text encoder (W2&Q2) :**
>
>    During training, we freeze the CLIP text encoder, following common practice in Text-to-Image models, which is because pretraining has already well-aligned the features in the latent space, and making the CLIP text encoder trainable would inevitably disrupt the general semantic structure, cause semantic drift, and lead to unstable training and difficulty in convergence. Thus, we choose to freeze the CLIP text encoder to maintain the stability of the semantic space, while learning a new corresponding token embedding for each emotion category to achieve emotional adaptability, thereby accomplishing the EICG task. To further investigate this, we also experiment with making CLIP text encoder trainable, but the results, shown in the table below, are suboptimal compared to freezing, demonstrating the higher effectiveness and practicality of our current setup. We will incorporate this discussion in the revised version.
>
>    | CLIP text encoder | FID ↓ | LPIPS ↑ | Emo-A ↑ | Sem-C ↑ | Sem-D ↑ |
>    | ----------------- | ----- | ------- | ------- | ------- | ------- |
>    | Trainable         | 43.87 | 0.709   | 75.98%  | 0.602   | 0.0311  |
>    | Frozen (Ours)     | 40.66 | 0.732   | 80.15%  | 0.641   | 0.0349  |

---

> > ### Author Response · Authors · 2025-11-28
> > **Response to Reviewer 2zyk (2/2)**
> >
> > 3. **About the diversity of generation (W3&Q3) :**
> >
> >    We would like to clarify that images in EmoSet express emotions across multiple dimensions, covering a wide range of scene types, object classes, and so on, rather than rigidly tying emotions to symbolic elements. For example, images conveying fear are not limited to terrifying faces but also include elements such as snakes, blood, deep corridors, dark rooms, and more.
> >
> >    In CoEmoGen, the captions further enrich the context of the attribute labels in EmoSet to achieve semantic coherence, the emotion-specific LoRA captures fine-grained exclusive elements of specific emotions, and inference involves randomly sampling visual embeddings from the corresponding emotion cluster, with none of these procedures amplifying bias. The qualitative visualizations and quantitative diversity metrics, including LPIPS (0.732) and Sem-D (0.0349), in our experiments clearly demonstrate CoEmoGen’s superiority in generating diverse content, showing that it does not adhere itself to symbolic and monotonous emotional expressions.
> >
> > 4. **About neutral phrases in captions (W4&Q3) :**
> >
> >    Indeed, neutral phrases do appear in sentence-level captions, but they do not occur in isolation and do not undermine the model’s ability to capture abstract emotional semantics; instead, they provide the physical basis for emotional expression, while our CoEmoGen learns to combine them with emotion-rich context (e.g., combining “mouth open” with “fire” to express fear, or with “laughter” to express amusement), representing the necessary transition from concrete to abstract. Our ablation analysis in Table 2 (a) clearly demonstrates this, showing that when the prompt does not include emotional priors serving to encourage the MLLM to generate captions focusing on emotion-related elements, that is, when the captions are dominated by neutral phrases, the resulting performance is suboptimal compared to their counterparts guided by emotional priors, highlighting the superiority of our emotion-rich captions and showcasing that CoEmoGen does not merely learn low-level visual co-occurrences.
> >
> > 5. **About readability (W5) :**
> >
> >    We will adjust the font sizes in the figures and tables to an appropriate level to improve the paper’s readability and clarity.

---

### Official Review · Reviewer_xnJi · 2025-10-31

**Soundness:** 3
**Presentation:** 3
**Contribution:** 3
**Rating:** 6
**Confidence:** 3

**Summary:**

This paper introduces CoEmoGen, a novel pipeline for Emotional Image Content Generation (EICG) that addresses the shortcomings of existing text-to-image and EICG models. The authors identify that prior methods, which rely on word-level attribute labels for guidance, suffer from semantic incoherence, ambiguity, and limited scalability. CoEmoGen tackles this by making two primary contributions: (1) It utilizes Multimodal Large Language Models (MLLMs) to generate context-rich, sentence-level captions that serve as semantically coherent guidance. (2) It proposes a Hierarchical Low-Rank Adaptation (HiLoRA) module, inspired by psychology, which decouples emotion modeling into polarity-shared LoRAs (for common low-level features) and emotion-specific LoRAs (for high-level semantics).

**Strengths:**

1. The paper correctly identifies a critical flaw in prior EICG work: the reliance on word-level attribute labels leads to semantic incoherence (e.g., unnatural "collage-like" images). The shift from isolated word-level guidance to sentence-level semantic guidance is a significant and logical paradigm shift that directly addresses this core problem, resulting in more natural and contextually sound images.

2. The design of the HiLoRA module is well-motivated by the psychological observation that emotions of the same polarity (e.g., positive/negative) share low-level visual features (like brightness) while differing in high-level semantics. This hierarchical decoupling is elegant and is shown to be effective via strong ablation studies, where removing either the polarity-shared or emotion-specific LoRAs leads to performance degradation.

**Weaknesses:**

1. The entire "coherent semantic acquisition" pipeline is fundamentally bottlenecked by the quality of the MLLM used for captioning. The authors acknowledge the risk of MLLM hallucinations and use a heuristic CLIP-based filtering method (discarding the bottom 20% ) to mitigate this. However, this filtering may not be robust enough to catch subtle semantic or emotional inaccuracies, and the model's performance is intrinsically tied to the chosen MLLM's capabilities.

2. While the creation of EmoArt is a strength in terms of scalability, its curation methodology appears flawed. The authors use a classifier pre-trained on EmoSet (natural images) to predict emotions for artistic images from WikiArt. This introduces a significant domain gap. A classifier trained on photos is unlikely to accurately capture the emotional expression in abstract, impressionist, or other non-photorealistic art styles, potentially biasing the resulting dataset.

**Questions:**

The choice of prompt is ablated, but the choice of MLLM is not. Were different MLLMs (e.g., LLaVA, GPT-5) experimented with for caption generation? How sensitive is the model's final performance (e.g., Emo-A or Sem-C) to the quality, verbosity, and style of the captions generated by different MLLMs?

---

> ### Author Response · Authors · 2025-11-28
> **Response to Reviewer xnJi**
>
> 1. **About choice of MLLM (W1&Q1) :**
>
>    During the process of coherent semantic acquisition, we by default use Qwen2-VL-7B to generate concise yet emotion-rich sentence-level captions as semantically coherent guidance. To further investigate how the choice of MLLM affects CoEmoGen’s EICG performance, we conduct an ablation comparison using InstructBLIP-7B, LLaVA-1.5-7B, Qwen2.5-VL-32B, and GPT-4o, each driven with the default prompt to generate the corresponding captions that are subsequently used as semantic guidance to train CoEmoGen. The results are shown in the table below.
>
>    | MLLM               | FID ↓ | LPIPS ↑ | Emo-A ↑ | Sem-C ↑ | Sem-D ↑ |
>    | ------------------ | ----- | ------- | ------- | ------- | ------- |
>    | InstructBLIP-7B    | 45.52 | 0.701   | 75.89%  | 0.610   | 0.0304  |
>    | LLaVA-1.5-7B       | 43.11 | 0.719   | 78.86%  | 0.624   | 0.0329  |
>    | Qwen2-VL-7B (Ours) | 40.66 | 0.732   | 80.15%  | 0.641   | 0.0349  |
>    | Qwen2.5-VL-32B     | 40.48 | 0.729   | 80.32%  | 0.636   | 0.0342  |
>    | GPT-4o             | 40.71 | 0.736   | 80.01%  | 0.639   | 0.0353  |
>
>    It can be observed that CoEmoGen’s performance is indeed influenced, to some extent, by the capabilities of the MLLM.  Specifically, employing captions generated by InstructBLIP-7B or LLaVA-1.5-7B leads to a performance drop, as their relatively weaker capabilities cause the captions to inevitably contain noise or hallucinations, resulting in lower reliability and less accurate guidance. However, in contrast, captions generated by the more powerful Qwen2.5-VL-32B and GPT-4o produce results that are not substantially different from the default Qwen2-VL-7B. We infer that this is because captioning itself is a relatively simple and not highly challenging task; once an MLLM reaches a certain level of capability, the quality of its generated captions becomes comparable, thus the choice of MLLM is no longer a key factor influencing CoEmoGen’s performance. In such cases, choosing a more lightweight and convenient MLLM offers a better trade-off between performance and efficiency. We will incorporate this discussion in the revised version.
>
>    Additionally, we also manually investigate the quality of the captions used by default. Specifically, we randomly sample 50 captions from each of the 8 emotion categories (400 samples in total) and invite three human annotators to individually rate them on linguistic accuracy and emotional authenticity using a five-point Likert scale (1 = poor, 5 = excellent). The statistical results of this evaluation are shown in the table below. It can be observed that over 87% of the samples receive high scores (≥4) in both aspects, while the proportion of low-scoring outlier samples remains below 5%. We also calculate the Intraclass Correlation Coefficient (ICC) among the human annotators, which is 0.93 (closer to 1 indicates better agreement), demonstrating a high level of inter-annotator agreement. These results further demonstrate that the captions currently employed are reliable and credible. We will also incorporate this discussion in the revised version.
>
>    | Score | Linguistic accuracy | Emotional authenticity |
>    | ----- | ------------------- | ---------------------- |
>    | 1     | 1.42%               | 1.92%                  |
>    | 2     | 2.08%               | 2.67%                  |
>    | 3     | 4.33%               | 8.25%                  |
>    | 4     | 28.08%              | 27.58%                 |
>    | 5     | 64.08%              | 59.58%                 |
>
> 1. **About EmoArt emotion label assignment (W2) :**
>
>    We acknowledge that using a pre-trained classifier to assign emotion labels to images in EmoArt does indeed introduce a domain gap. We would like to point out, however, that this design is essentially a pragmatic choice intended to demonstrate our standardized construction paradigm, allowing users, at minimal cost, to customize diverse emotion-rich datasets and generate emotionally stimulating images tailored to their specific needs, thereby showcasing the scalability of CoEmoGen. We also apply an emotion confidence threshold to filter out samples with clear emotional expression and prevent ambiguity, to alleviate this limitation. The visualizations in Figure 7 demonstrate the feasibility of this paradigm.
>
>    Furthermore, if higher demands are required, users can also manually annotate a small batch of samples corresponding to their specific needs, which does not incur significant additional cost, to fine-tune the pre-trained classifier, enabling it to acquire the necessary domain-specific knowledge to reduce the gap, thereby producing more precise emotion labels.

---

### Official Review · Reviewer_QKK7 · 2025-11-01

**Soundness:** 3
**Presentation:** 3
**Contribution:** 2
**Rating:** 6
**Confidence:** 5

**Summary:**

The work aims to address two core challenges in the field: first, the difficulty general text-to-image models face in accurately capturing and generating abstract emotional concepts like 'awe' or 'contentment'; and second, the limitations of existing EICG-specific model (EmoGen) that rely on word-level attribute labels for guidance. This reliance not only leads to semantically incoherent results (e.g., illogical combinations of elements) and ambiguous emotional expression.

To overcome these challenges, CoEmoGen proposes a two-pronged solution. First, it leverages Multimodal Large Language Models (MLLMs) to automatically generate context-rich sentence-level captions, replacing the flawed word-level labels to achieve more coherent and rich semantic guidance. Second, inspired by psychological observations, it designs a novel Hierarchical LoRA (HiLoRA) module. This module refines emotion modeling by using "polarity-shared LoRAs" (to capture common features of positive/negative emotions) and "emotion-specific LoRAs" (to capture the unique semantics of each emotion).

According to the experiments, the advantages of this method are the improvements in both semantic coherence (producing more natural and logical images) and emotional faithfulness. Furthermore, its standardized data construction pipeline (using MLLMs) is proven to be highly scalable, allowing it to be effortlessly applied to entirely new domains, such as artistic paintings—a task.

**Strengths:**

A substantive assessment of the strengths of the paper, touching on each of the following dimensions: originality, quality, clarity, and significance. We encourage reviewers to be broad in their definitions of originality and significance. For example, originality may arise from a new definition or problem formulation, creative combinations of existing ideas, application to a new domain, or removing limitations from prior results.

The paper begins by defining a clear objective: precisely identifying the core weaknesses of current EICG methods, namely their semantic incoherence and poor scalability.

Finally, the claims are substantiated by comprehensive and well-designed experiments. The authors go beyond standard quantitative and qualitative comparisons by including crucial ablation studies to justify their architectural choices, a well-executed user study to validate the perceptually-driven claim of "semantic coherence," and a practical demonstration (the EmoArt dataset) to prove the tangible benefits of their scalable pipeline.

**Weaknesses:**

CoEmoGen doesn't completely escape label dependency. It merely replaces 'fine-grained attribute labels' with a 'coarse-grained emotion label'. Therefore, its scalability is relative; it still requires a dataset pre-annotated with emotions as a starting point, rather than being able to learn from completely unsupervised images.

**Questions:**

**Question 1 (On the Definition and Focus of EICG):**

How exactly do the authors define Emotional Image Content Generation (EICG)? The examples provided in the paper (e.g., the word clouds figure) suggest that 'emotional images' encompass at least two distinct categories:

1.  Images that **evoke** a specific emotion *in the viewer* (e.g., serene landscapes for 'Awe').
2.  Images that **depict** a subject *expressing* an emotion (e.g., a person or animal 'showing teeth' for 'Anger').

Does the CoEmoGen framework demonstrate a particular focus or preference for one of these aspects (evocation or expression), or does it treat both as equally valid means of achieving an "emotionally faithful" generation?

**Question 2 (Critique on the Novelty and Scalability of the MLLM Pipeline):**

The paper presents its MLLM-based pipeline for generating 'sentence-level captions' as a key advantage for scalability, positioning it as a superior alternative to manual 'word-level labels.' However, the MLLM prompt itself still requires a high-level `emotion` label as a prior. This suggests the method is not fully independent of labels.

This leads to two concerns:

1.  Could this same "label-to-caption" generation process also be applied to the EmoGen baseline with minor adaption, like 'label-to-more-labels'? If so, this would imply the novelty lies in the *guidance signal* (captions) rather than the *pipeline's architecture*.
2.  Furthermore, the pipeline of using a few high-level labels (like an emotion category) to prompt an LLM to generate richer, descriptive captions is an increasingly common practice in the generative field. Does this component of CoEmoGen truly represent a significant and novel advancement in scalability, or is it more of an effective application of an existing paradigm?

**Details Of Ethics Concerns:**

As stated in the paper, EmoArt was collected from WikiArt, and the authors seem to have plans to open-source it. Please check the copyright status.

---

> ### Author Response · Authors · 2025-11-28
> **Response to Reviewer QKK7 (1/2)**
>
> 1. **About label dependency (W1) :**
>
>    We acknowledge that CoEmoGen indeed requires emotion labels as a starting point. However, compared with EmoGen, which not only requires emotion labels but also relies on word-level attribute labels that incur high annotation costs and suffer from potential issues such as lack of contextual association, weak correlation to emotions, and missing annotations, CoEmoGen operates solely with emotion labels, minimizing annotation requirements and laying the foundation for high scalability. Furthermore, when aiming to extend to unlabeled images, a pre-trained emotion classifier can be used in combination with a confidence threshold to assign relatively reliable emotion labels to new images at minimal cost, enabling seamless integration into our pipeline. Our automated standardized construction paradigm allows users to effortlessly customize diverse emotion-rich datasets and generate emotionally stimulating images tailored to their specific needs, with the construction of EmoArt in Section 4.4 serving as an intuitive example, clearly demonstrating CoEmoGen’s high flexibility and scalability.
>
> 2. **About definition and focus of EICG (Q1) :**
>
>    We define Emotional Image Content Generation (EICG) as the reverse process of Visual Emotion Analysis (VEA), that is, generating semantically clear and emotionally faithful visual content to convey a given emotion category, consistent with EmoGen. We adopt the most commonly employed Mikels model [R1], which classifies emotions into eight categories.
>
>    As you accurately observed, the conveyance of emotional imagery generally manifests in two forms: evocative, where emotion is induced through elements such as scene, tone, or composition, and expressive, where emotion is directly communicated through facial expressions or body language of humans or animals. CoEmoGen is able to accommodate both forms, as they represent complementary and equally valid pathways toward achieving emotionally faithful generation, and accordingly does not favor either mode but instead aims to support both within a unified paradigm. We will include this clarification in the revised version.
>
>    [R1] Mikels, Joseph A., et al. "Emotional category data on images from the International Affective Picture System." Behavior research methods 37.4 (2005): 626-630.

---

> ### Author Response · Authors · 2025-11-28
> **Response to Reviewer QKK7 (2/2)**
>
> 3. **About introducing more labels for EmoGen (Q2-1) :**
>
>    Due to the inherent limitation that the attribute loss in EmoGen is optimized in a word-level space, expanding more labels may not necessarily yield benefits. To empirically investigate this, we extract key phrases from our generated captions and use them as expanded “more labels” to guide EmoGen, conducting an ablation analysis on this variant. The results are shown in the table below.
>
>    | Method                  | FID ↓ | LPIPS ↑ | Emo-A ↑ | Sem-C ↑ | Sem-D ↑ |
>    | ----------------------- | ----- | ------- | ------- | ------- | ------- |
>    | EmoGen                  | 41.60 | 0.717   | 76.25%  | 0.633   | 0.0335  |
>    | EmoGen with more labels | 43.79 | 0.691   | 73.54%  | 0.622   | 0.0293  |
>    | CoEmoGen (Ours)         | 40.66 | 0.732   | 80.15%  | 0.641   | 0.0349  |
>
>    It can be observed that although the more labels introduce additional information, they do not lead to improvements; instead, performance drops. This is because introducing more labels further exacerbates semantic fragmentation, making the optimization direction more unstable, thereby underscoring EmoGen’s flaw of relying on context-lacking and incoherent word-level labels.
>
>    In contrast, our CoEmoGen leverages semantically coherent and context-rich captions as guidance, optimized through a semantic loss in the sentence-level space and integrated with the psychology-inspired HiLoRA structure, forming a cohesive whole, demonstrating superior performance on the EICG task. We will incorporate this discussion in the revised version.
>
>
> 4. **About scalability (Q2-2) :**
>
>    Distinguishing from existing practices, in the process of coherent semantic acquisition, we employ an emotion-focused, guided caption generation strategy, using carefully designed prompts to direct the MLLM to produce captions that are both concise and emotion-rich. Unlike static or single-step pipelines, our standardized construction paradigm can first assign labels to unannotated images using a pre-trained classifier with a confidence threshold, and then, with the assistance of the MLLM, generate emotion-rich captions that are subsequently used to train CoEmoGen integrated with the psychology-inspired HiLoRA. This fully automated, end-to-end workflow allows users to customize diverse emotion-rich datasets and generate emotionally stimulating images tailored to their specific needs at minimal cost, highlighting CoEmoGen’s high flexibility and scalability, with the construction of EmoArt serving as a highly representative example.
>
> 5. **About copyright concerns (Ethics Concerns) :**
>
>    Thank you for your reminder! The images in EmoArt are sourced from the WikiArt website and, according to the relevant terms and policies, are permitted for non-commercial research use. Upon release, we will provide only download links rather than redistributing any original images directly, along with the corresponding derived content in EmoArt, including emotion and caption annotations, and model checkpoints. All access will be controlled via a detailed data usage agreement, which restricts usage to non-commercial research purposes.

---

### Author Response · Authors · 2025-12-03
**Summary of Reviews and Rebuttal (1/2)**

Dear PCs, SACs, ACs, and Reviewers,

We sincerely thank you for your continuous time and effort throughout the paper review process, and especially appreciate the four reviewers (QKK7, xnJi, 2zyk, LzoM) for their constructive feedback, which has helped us further strengthen the quality of our paper. Given the unexpected changes in the review process and the approaching deadline, we would like to take this opportunity to provide a brief summary and clarification of the reviews and our corresponding responses to facilitate your final assessment.

---

We are deeply grateful to the reviewers for recognizing the strengths of our paper in their reviews, including:

- The **clarity of our motivation**, precisely identifying the core flaw of current EICG work, namely **semantic incoherence and poor scalability** (QKK7: Strength 1; xnJi: Strength 1);
- The **innovation and significance of the CoEmoGen pipeline**, shifting from isolated word-level guidance to **sentence-level semantically coherent guidance** (xnJi: Summary & Strength 1);
- The **novelty of our HiLoRA module** (QKK7: Summary; LzoM: Strength 1), inspired by psychological observations, and its **elegance** (xnJi: Strength 2; 2zyk: Strength 2);
- The **high scalability of CoEmoGen**, demonstrated by the construction of the **EmoArt dataset** (QKK7: Summary & Strength 2; 2zyk: Strength 1 & Strength 3);
- The **comprehensive and well-designed experiments (qualitative, quantitative, user studies, and ablation analyses)** (QKK7: Strength 2) showing the **superiority and effectiveness of CoEmoGen** (QKK7: Summary & Strength 2; xnJi: Strength 2; 2zyk: Summary; LzoM: Strength 2).

---

Meanwhile, we have carefully provided point-by-point responses to the reviewers’ questions and concerns in the rebuttal, which we summarize below.

**Reviewer QKK7** raised questions regarding **label dependency** (Weakness 1), **EICG definition** (Question 1), **introducing more labels for EmoGen** (Question 2-1), **scalability** (Question 2-2), and **WikiArt copyright** (Ethics Concerns). We responded as follows:

- Explained that while CoEmoGen requires emotion labels as a starting point, it fully obviates the need for costly word-level attributes used by EmoGen, and a pre-trained emotion classifier with a confidence threshold can effortlessly extend it to unlabeled images;
- Clarified that emotional imagery in EICG generally manifests in two forms, evocative and expressive, and CoEmoGen does not favor either;
- Conducted an ablation of the EmoGen-with-more-labels variant, observing performance drops, highlighting limitations of EmoGen’s word-level optimization and the superiority of CoEmoGen’s semantically coherent guidance;
- Explained that our standardized construction paradigm is uniquely scalable, using emotion-focused captioning and a fully automated workflow for effortless customization of emotion-rich datasets;
- Addressed copyright concerns by confirming WikiArt images are allowed for non-commercial research, and committed that our release provides only original download links, not redistributed images.

---

> ### Author Response · Authors · 2025-12-03
> **Summary of Reviews and Rebuttal (2/2)**
>
> **Reviewer xnJi** raised questions regarding the **choice of MLLM** (Weakness 1 & Question 1) and **EmoArt emotion label assignment** (Weakness 2). We responded as follows:
>
> - Conducted comprehensive ablations with multiple MLLMs (InstructBLIP-7B, LLaVA-1.5-7B, Qwen2-VL-7B, Qwen2.5-VL-32B, GPT-4o), showing that once an MLLM reaches sufficient capability, caption quality is comparable, making MLLM choice no longer a key factor for CoEmoGen’s performance, with lightweight models offering better efficiency trade-offs;
> - Performed manual evaluation (5-point scale) by three human annotators on 400 randomly sampled captions, showing over 87% received high scores (≥4) for linguistic accuracy and emotional authenticity, with high inter-annotator agreement (ICC = 0.93), demonstrating the reliability of the captions used;
> - Explained that the current emotion label assignment is a minimal-cost, pragmatic choice to demonstrate CoEmoGen’s flexibility, with Figure 7 showing its feasibility, and provided alternative solutions for higher-demand scenarios without significant additional cost.
>
> **Reviewer 2zyk** raised questions regarding the **quality of MLLM-generated captions** (Weakness 1 & Question 1), **freezing of CLIP text encoder** (Weakness 2 & Question 2), **diversity of generation** (Weakness 3 & Question 3), and **neutral phrases in captions** (Weakness 4 & Question 3). We responded as follows:
>
> - Provided detailed human annotator evaluation for caption quality (same as response to xnJi), demonstrating reliability without manual intervention;
> - Explained that freezing the CLIP text encoder follows standard practice to maintain semantic stability, with ablation validation attempts showing that a trainable encoder indeed led to worse performance;
> - Clarified that our training corpus is diverse and generation shows emotional expression across multiple dimensions rather than symbolic representations, with qualitative visualizations and quantitative diversity metrics (LPIPS = 0.732, Sem-D = 0.0349) confirming content variety;
> - Emphasized that neutral phrases in our emotion-rich captions provide a necessary physical basis for emotional expression and are not isolated, but properly contextualized with emotion-rich elements, thereby preventing the model from merely learning low-level visual co-occurrences, as validated by our ablation analysis in Table 2 (a).
>
> **Reviewer LzoM** raised questions regarding the **hierarchy in HiLoRA** (Weakness 1), **potential LLM+T2I baseline** (Weakness 2), **missing related work** (Weakness 3), **user study scale** (Question 1), and the **implementation of targeted emotion transfer** (Question 2). We responded as follows:
>
> - Clarified that "hierarchy" in HiLoRA refers to psychological emotional cognition (polarity and discrete emotions) rather than network architecture, and provided the requested ablation showing that a sequential design underperformed our original parallel fusion;
> - Conducted experiments with four LLM+T2I combinations, demonstrating that even the strongest combination (GPT-5+FLUX.1) underperformed CoEmoGen, confirming the necessity of employing the dedicated method CoEmoGen to effectively accomplish EICG.
> - Acknowledged the imprecise description of EmoArt, committed to narrowing the wording and incorporating necessary discussion of the ArtEmis dataset family;
> - Detailed that the user study evaluated 1,024 images total (16 images × 2 groups × 4 methods × 8 emotions);
> - Explained that the targeted emotion transfer is achieved by concatenating emotion descriptors with neutral element embeddings while activating the corresponding LoRAs, enabling coherent emotional fusion.
>
> ---
>
> We greatly regret not having the opportunity to discuss with the reviewers, but we have provided comprehensive responses to all raised issues, which we believe largely resolve the concerns.
>
> We hope this summary helps you gain a clearer grasp of the reviews and our responses. We will continue to work diligently to incorporate all suggestions and improvements in the final version.
>
> Thank you once again for your time and consideration.
>
> Sincerely,
>
> Authors of Paper 13488

---

### Meta-Review · Area_Chair_UW7h · 2026-01-06

**Summary:**

This paper proposes CoEmoGen, a novel pipeline for Emotional Image Content Generation that addresses semantic incoherence and limited scalability in existing methods by leveraging Multimodal Large Language Models to produce sentence-level captions for richer semantic guidance and introducing a Hierarchical LoRA module inspired by psychological insights to model both polarity-shared and emotion-specific features. The initial reviews generally acknowledge the paper's clear motivation, innovative approach, and comprehensive experiments, but raise concerns regarding label dependency, the quality and validation of MLLM-generated captions, potential biases in the curated EmoArt dataset, the true hierarchical design of HiLoRA, and the need for additional baselines (e.g., LLM+T2I combinations). Reviewers also note issues with readability and suggest comparisons with related emotional art datasets.

**Reviewer Concerns:**

In the rebuttal, the authors addressed reviewers' concerns by clarifying the definition and scope of Emotional Image Content Generation, conducting ablation studies to validate the superiority of their method over variants with more labels or sequential HiLoRA designs, and performing experiments with multiple MLLMs supported by human evaluation to demonstrate caption reliability. They justified the frozen CLIP text encoder with comparative results, added a new LLM+T2I baseline showing CoEmoGen's advantage, detailed the user study scale and emotion transfer mechanism, and committed to improving readability and refining the EmoArt dataset description while addressing copyright issues. However, the fundamental reliance on initial emotion labels persists, meaning the method is not fully unsupervised and its scalability remains relative. Potential domain gaps in label assignment for artistic images and inherent biases from the training data are acknowledged but not fully resolved.

**Reviewer Scores:**

If the reviewers could fully participate in the discussion, they might adjust their ratings upward after engaging with the authors' detailed rebuttal, particularly appreciating the comprehensive ablation studies, the human evaluation of MLLM captions, and the direct comparison against a new LLM+T2I baseline, which collectively strengthen the paper's empirical validation and novelty claims. Reviewers who questioned the fundamental label dependency and the non-hierarchical nature of HiLoRA might maintain their critical stance, as the rebuttal acknowledges but does not solve these core conceptual limitations. The discussion could lead to a consensus that the paper presents a solid, well-evaluated engineering solution with clear practical benefits, especially in scalability and coherence, but its theoretical advancement remains incremental, potentially resulting in a final rating that is positive yet cautious, around the acceptance threshold.

---

### Decision · Program_Chairs · 2026-01-26

Accept (Poster)